# GEOMETRY-AWARE EUCLIDEAN DIFFUSION LANGUAGE GENERATION

## ABSTRACT

We formulate a powerful generative framework for language, premised upon modeling discrete token generation as continuous trajectories of a Gaussian stochastic process in a Euclidean space. Specifically, to address the challenge of the high-dimensional discrete nature inherent in language data, we devise two core components: a projection function to embed discrete tokens into a continuous domain and a metric function to infer the conditional probability distribution of subsequent tokens from continuous embeddings. Subsequently, we employ a forward diffusion process that incrementally perturbs the data distribution towards a tractable standard Gaussian prior. To learn the generative reverse process, we formulate a novel *data geometry-aware score* that explicitly exploits the inherent manifold structure of the discrete language data to refine the fidelity of the score approximation. Since direct optimization of the score function is intractable, we propose optimizing a tractable surrogate objective, the Relaxed Evidence Lower Bound, which ensures a bounded approximation error via continuous relaxation. Finally, we critically reassess conventional evaluation protocols and introduce a novel comprehension score, designed to enable a more robust and equitable performance comparison against competing architectures. Empirical validation on the LM1B and OpenWebText benchmarks corroborates the effectiveness of our proposed framework. *The proposed method significantly outperforms state-of-the-art discrete diffusion and autoregressive schemes, indicating a very promising direction for language modeling.*

## 1 INTRODUCTION

The landscape of natural language processing has been profoundly reshaped by the advent of large language models (LLMs) (Radford et al., 2018; Achiam et al., 2023; Team et al., 2023; Liu et al., 2024), predominantly architected as auto-regressive systems based on the Transformer framework. These models, such as the GPT series, generate text sequentially, predicting one discrete token at a time conditioned on the preceding sequence. This paradigm has demonstrated remarkable performance across a spectrum of tasks, from text generation and translation to complex reasoning. Despite their success, auto-regressive models (Tsay, 1989; Krolzig, 1997) are inherently constrained by their sequential, token-by-token generation process, which can lead to issues like error propagation and limited flexibility in editing or refining generated text.

Concurrently, a distinct class of generative models, known as diffusion models (Sohl-Dickstein et al., 2015; Ho et al., 2020; Song et al., 2020b), has emerged as the state-of-the-art in continuous domains like image (Rombach et al., 2022), video (Ho et al., 2022), and audio synthesis (Kong et al., 2020). These models operate through a dual-process mechanism: a forward noising process that incrementally perturbs data with noise until pure noise, and a reverse denoising process where a neural network learns to iteratively remove this noise to recover the original data structure. The power of this approach lies in its ability to transform a complex generation task into a sequence of well-defined denoising steps (Ho et al., 2020), yielding outputs of exceptionally high fidelity.

The success of diffusion models in continuous domains has inspired their application to the discrete realm of natural language (Lou et al., 2023; He et al., 2022; Sahoo et al., 2024), leading to the development of diffusion language models. These models bridge the gap by establishing a similar probability flow for discrete variables (Meng et al., 2022; Sun et al., 2022; Austin et al., 2021) using the probability transition process and then applying the discrete/concrete score-matching framework to realize the reverse diffusion generation. Instead of discrete diffusion, there are some approaches

Table 1: Comparison of different diffusion language (or discrete data) generation methods: D3PM (Austin et al., 2021), CSM (Meng et al., 2022), SEDD (Lou et al., 2023), Plaid (Gulrajani & Hashimoto, 2023), RDLM (Jo & Hwang, 2025), and Ours.

| Methods | D3PM | CSM | SEDD | Plaid | RDLM | Ours |
|---|---|---|---|---|---|---|
| Forward | $P(\mathbf{x}_t) = \bar{\mathbf{Q}}_t P(\mathbf{x})$ | | | $\mathbf{x}_t = \alpha(t)\mathbf{x}_0 + \beta(t)\mathbf{n}$ | $\widetilde{\mathbf{x}}_t = \alpha(t)\widetilde{\mathbf{x}}_0 + \beta(t)\widetilde{\mathbf{n}}$ | $\mathbf{x}_t = \alpha(t)\widetilde{\mathbf{x}}_0 + \beta(t)\mathbf{n}$ |
| State space | Discrete points (Complete graph) | | | Euclid. | Riemann. | Euclid. + Riemann.[3] |
| Objective | KL | $\frac{P(\mathbf{x}_\mathcal{N}) - P(\mathbf{x})}{P(\mathbf{x})}$ [1] | $\frac{P(\mathbf{x}_\mathcal{N})}{P(\mathbf{x})}$ | Continuous | KL | RELBO |
| Geo-aware | ✓ | ✓ | ✓ | ✗ | ✓ | ✓ |

[1] $\mathbf{x}_N$ indicates all the neighbors of the sample $\mathbf{x}$. For language generation, it indicates all other different classes than $\mathbf{x}$.

[2] "~" operator in this table indicates that the corresponding variable is on the Non-Euclidean data manifold structure. $\mathbf{x}_0$, $\mathbf{x}_t$, and $\mathbf{n}$ represent the cleaned sample, interpolated state, and Gaussian noise, respectively. $\alpha(t)$ and $\beta(t)$ correspond to their respective weights.

[3] "Euclid. + Riemann." indicates that the score function of the proposed method can master the exact data manifold structure ($\mathbf{x}_0$ is exactly on manifold) but diffusion in the Euclidean space.

that try to process the discrete variable in an explicitly continuous manner, e.g., latent diffusion (Gulrajani & Hashimoto, 2023), or Riemannian diffusion (Jo & Hwang, 2025). The primary benefits of these diffusion approaches generally lie in the *flexibility from bidirectional attention* and *continuous-space reasoning with continuous-time probability flow*. Unlike autoregressive models that are committed to a fixed prefix, diffusion models refine the entire sequence representation simultaneously. This allows for non-monotonic generation, where the model can fill in, edit, and refine text from a holistic "draft," offering unprecedented control over the generative process and potentially capturing more nuanced semantic relationships in the continuous latent space. Moreover, the inherent continuous probability flow eliminates the limitations of computational resources during the generation process, i.e., allowing more inference steps to achieve more accurate results (Lou et al., 2023). While discrete and continuous diffusion models are typically designed to process their respective data types, *a continuous model for discrete language data is particularly promising*. This is because discrete diffusion models are fundamentally incapable of processing continuous data in a lossless manner; thus, if a continuous model can effectively process discrete data, it would emerge as a potent unified framework for multi-modal data processing. Moreover, continuous diffusion models operate within a continuous state space containing infinite states, allowing them to better capture subtle distinctions during the diffusion process. Furthermore, the continuous language diffusion approaches establish continuous trajectories in both sample and probabilistic spaces, which can leverage the previous mature image diffusion methods (Sauer et al., 2024; Song et al., 2023; Zhu et al., 2025) to boost performance and efficiency. However, this novel approach is not without its challenges. A significant difficulty of current diffusion language models is the construction of a reasonable and effective score-matching mechanism. Due to the discrete nature of language data, classic continuous score matching (Song et al., 2020b) is difficult to apply directly to the discrete data manifold.

In this paper, we propose a geometry-aware score-matching technique to tackle the issue of continuous diffusion models for discrete language data, achieving Gaussian diffusion language generation with awareness of the intrinsic discrete data structure. Moreover, to enable score learning, we derive a learnable Relaxed Evidence Lower Bound. In summary, the main contributions of this work are:

- we formulate a geometry-aware score function for discrete language processing in continuous space, and derive a relaxed evidence lower bound to optimize the score;
- we theoretically analyze the connection between geometry-aware score matching and traditional score-matching paradigms; and
- we propose a novel rank-based metric using an LLM to evaluate the generation performance and conduct extensive experiments on LM1B and OpenWebText benchmarks to validate the effectiveness of the proposed method.

## 2 PRELIMINARY

**Continuous Data Synthesis via Euclidean Diffusion**. Through progressively perturbing data via a tractable distribution, e.g., Standard Gaussian, we can derive the forward trajectories (stochastic differential equations) (Song et al., 2020b) that link the data and the prior distribution as

$$d\mathbf{x}_t = f(\mathbf{x}_t, t)dt + g(t)d\mathcal{B}_t, \tag{1}$$

where $f(\cdot, t)$ and $g(t)$ indicate the drift and diffusion coefficient functions, respectively, $t \in [0, 1]$ represents the timestamp, $d\mathcal{B}_t$ indicates a standard Brownian motion, and $\mathbf{x}_t$ is a continuous state that is on the trajectories between data and noise distributions. Moreover, due to the isotropy of Euclidean space, such a forward process has an analytical solution, which is given by $q(\mathbf{x}_t|\mathbf{x}_0) = \mathcal{N}(\alpha_t\mathbf{x}_0, \sigma_t^2\mathbf{I})$. Then, the signal-to-noise ratio of this process can be formulated as $\mathrm{SNR}(t) =$

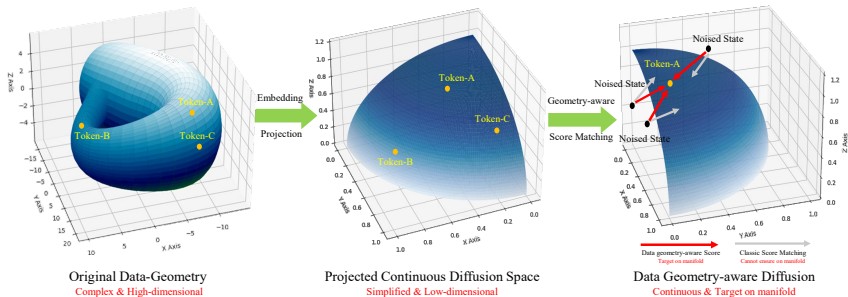

Figure 1: Illustration of our data geometry-aware diffusion process. We first utilize an embedding projection $\mathcal{P}^{(c)}(\cdot)$ to transfer the original discrete and complex data geometry into a continuous and low-dimensional diffusion space. Based on this flatter space, we further introduce a data geometry-score matching technique to achieve language generation from Gaussian noise. Note that the proposed diffusion scheme can *rigorously* ensure each diffusion denoising step points to the on-manifold data token points, which *cannot* be achieved by the classic manner of continuous diffusion.

$\alpha_t^2/\sigma_t^2$, which can represent the transition between the data and noise. Moreover, recent work also indicates that a smooth transition, marked by linear or smooth curves of $\text{SNR}(t)$, can benefit the generation quality of diffusion model (Kingma et al., 2021). Then, with linear drifting and diffusion functions, we can reverse this process in the data generator, which progressively removes the noise to derive clean data as

$$d\mathbf{x} = [f(\mathbf{x}_t, t) - g(t)^2 \nabla_\mathbf{x} \log(p_t(\mathbf{x}))]dt + g(t)d\widetilde{\mathcal{B}}_t; \quad p_t(\mathbf{x}_t) = \mathcal{N}(\alpha_t\widehat{\mathbf{x}}_0, \sigma_t^2\mathbf{I}), \tag{2}$$

where $\widehat{\mathbf{x}}_0$ is estimated by the neural network from $\mathbf{x}_t$.

**Diffusion language models** can be generally divided into *discrete* (Lou et al., 2023; Sahoo et al., 2024) or *continuous* (Jo & Hwang, 2025), depending on the modeling approach. Specifically, the *discrete language diffusion models*, also known as the masked diffusion models (Naveed et al., 2025), define the forward perturbation process that gradually transfers the prior data distribution to the absorbing state (or uniform distribution) as $q(\mathbf{x}_t|\mathbf{x}) = \text{Cat}(\mathbf{x}_t|\bar{\mathbf{Q}}_t\mathbf{x}) = \text{Cat}(\mathbf{x}_t; \mathbf{Q}_t \cdot \mathbf{Q}_{t-1} \cdots \mathbf{Q}_1\mathbf{x})$, where $\text{Cat}(\cdot)$ indicates the categorical distribution over the one-hot vector $\mathbf{x}$, and $\mathbf{Q}_t$ defines the transition matrix from $\mathbf{x}_{t-1}$ to $\mathbf{x}_t$. The backward probability flow is formulated as

$$q(\mathbf{x}_{t-1}|\mathbf{x}_t, \mathbf{x}_0) = \text{Cat}(\mathbf{x}_{t-1}; p = \frac{\mathbf{x}_t\mathbf{Q}_t^\mathsf{T} \odot \mathbf{x}_0\bar{\mathbf{Q}}_{t-1}}{\mathbf{x}_0\bar{\mathbf{Q}}_t\mathbf{x}_t^\mathsf{T}}), \tag{3}$$

where $\mathbf{x}_0$ is estimated through a learnable neural network with $\mathbf{x}_t$ sampled from $q(\mathbf{x}_t|\mathbf{x}_{t+1}, \mathbf{x}_0)$.

Moreover, the *continuous language diffusion models* (Jo & Hwang, 2025; Gulrajani & Hashimoto, 2023) were designed based on direct synthesis of the language within a continuous space. RDLM (Jo & Hwang, 2025) relaxes the token state space onto a hyper-sphere ($\|\mathbf{x}\|_2 \equiv 1$), and introduces a logarithm bridge process based on a stochastic differential geometric process $d\mathbf{x}_t^k = \gamma_t \frac{\phi_t(\mathbf{e}_k - \cos\phi_t\mathbf{x}_t^k)}{\sin\phi_t}dt + \sigma_t d\mathcal{B}_t$ on the hyper-sphere. Plaid (Gulrajani & Hashimoto, 2023) introduces latent diffusion into language generation that directly transfers the discrete language into a continuous Euclidean space and applies a standard continuous diffusion model as Eqs. (1) and (2). Moreover, Table 1 summarizes the differences between different methods.

## 3 PROPOSED METHOD

Continuous diffusion models progressively perturb target data to a prior distribution, using continuous flows in both probability and sample space, e.g., Gaussian in Euclidean (Ho et al., 2020), or von Mises–Fisher on the sphere (Dosi et al., 2025), and then reverse the perturbation trajectories to synthesize data from noise. However, for discrete data such as language tokens, building practical end-to-end continuous diffusion language generation models presents two main challenges. The first is the high dimensionality induced by large vocabulary sizes, which makes it computationally intractable to process raw data. The second is the inherent discreteness of data geometry, which renders the differential operator in the traditional score function, $\nabla_\mathbf{z} \log p(\mathbf{z})$, mathematically undefined, as discrete variables lack the continuous neighborhood required for differentiation. To address these issues, we construct a diffusion-based generation process within a projected flat embedding space with the corresponding geometric-aware score matching, as illustrated in Fig. 1. In what follows, we will detail our method.

### 3.1 DATA GEOMETRY-AWARE SCORE-MATCHING IN EUCLIDEAN SPACE

To enable discrete variable generation from continuous trajectories, we must explore a continuous space for generation and the corresponding projection operators between the discrete and continuous spaces. We first define the continuous projection operator $\mathcal{P}^{(c)}(\cdot): \{0,1\}^g \to \mathbb{R}^s (g \gg s)$, where $\{0,1\}^g$ indicates a $g$-dimension one-hot vector), e.g., $\mathbf{x}_0 = \mathbf{w}\mathbf{z} + \boldsymbol{\epsilon}^1$ with $\mathbf{w} \in \mathbb{R}^{s \times g}$ being an embedding dictionary or a linear layer, $\mathbf{x}_0 \in \mathbb{R}^s$ (resp. $\mathbf{z} \in \{0,1\}^g$) represents the continuous (resp. discrete) representation of a text token, and $\boldsymbol{\epsilon}$ is multivariate Gaussian (been skipped). We further introduce its reverse probabilistic metric function $\mathcal{P}^{(r)}(\cdot) : \mathbb{R}^s \to \{0,1\}^g$, ensuring $\mathbf{z} = \mathcal{P}^{(r)}(\mathbf{x}_0)$. In what follows, we give the derivation of a continuous diffusion process for discrete variables based on the modeling of a surrogate probability flow.

We first reparameterize the intractable probability flow $p(\mathbf{z}_{t_i}|\mathbf{z}_{t_{i+1}})$ of discrete distribution $p(\mathbf{z})$, using the *continuous surrogate flow* $p(\mathcal{P}^{(r)}(\mathbf{x}_{t_i})|\mathcal{P}^{(r)}(\mathbf{x}_{t_{i+1}}))$ , abbreviated as $p(\mathbf{x}_{t_i}|\mathbf{x}_{t_{i+1}})$ , where $\mathbf{x}_t$ evolves by a diffusion process of $\mathbf{x}_t = \alpha(t)\mathbf{x}_0 + \beta(t)\mathbf{n}$ (Ho et al., 2020). Then, a continuous diffusion process for a discrete variable can be formulated as the following discrete-time probability flow:

$$p(\mathbf{z}) = \int \left[ p(\mathbf{z}|\mathbf{x}_{t_0}) \prod_{i=0}^{N-1} p(\mathbf{x}_{t_i}|\mathbf{x}_{t_{i+1}}) \ p(\mathbf{x}_{t_N}) \right] d\mathbf{x}_{t_N}, \tag{4}$$

where $0 = t_0 < t_1 < \cdots < t_N = 1$; $p(\mathbf{z})$ is a vector of distribution probability for all cases of discrete variable $\mathbf{z}$; $p(\mathbf{x}_t)$ is the probabilistic density of variable $\mathbf{x}_t$; and $\mathbf{x}_{t_N}$ represent the sample of prior distribution of diffusion process (Gaussian). As $|t_i - t_{i+1}| \to 0$, we can also formulate the following continuous-time probability flow:

$$p(\mathbf{z}) = \int \left\{ \underbrace{p(\mathbf{z}|\mathbf{x}_0)}_{(1^{st})} \ [ \underbrace{\int_1^0 \frac{dp(\mathbf{x}_t)}{dt} dt}_{(2^{nd})} + \underbrace{p(\mathbf{x}_1)}_{(3^{rd})} ] \right\} d\mathbf{x}_1, \tag{5}$$

where the $1^{st}$ term represents the condition probability of the step of discretization; the $2^{nd}$ term indicates the integral of reverse probability transition of diffusion process; the $3^{rd}$ term denotes the density of prior Gaussian sample $\mathbf{x}_1$; and $\mathbf{x}_t$ evolves with Eq. (1). Since that the prior Gaussian variable $\mathbf{x}_1$ can be easily and strictly sampled, to approximate the $p(\mathbf{z})$ with $p_{\boldsymbol{\theta}}(\mathbf{z})$, the critical challenge lies in the approximation of $p(\mathbf{z}|\mathbf{x}_0)$ and $dp(\mathbf{x}_t)/dt$ from terms ($1^{st}$) and ($2^{nd}$), respectively. Based on previous works (Risken, 1989; Song et al., 2020b), we can regularize the approximation process as

$$\mathcal{L}_t = \begin{cases} \mathcal{D}_0\left(p_{\boldsymbol{\theta}}(\hat{\mathbf{z}}|\mathbf{x}_t), p(\mathbf{z}|\mathbf{x}_t)\right), & t = 0; \\ \mathcal{D}_1\left(\frac{dp_{\boldsymbol{\theta}}(\mathbf{x}_t)}{dt}, -\frac{d(f(\mathbf{x}_t,t)p(\mathbf{x}_t))}{d\mathbf{x}_t} + \frac{1}{2}\frac{d^2(g^2(\mathbf{x}_t,t)p(\mathbf{x}_t))}{d\mathbf{x}_t^2}\right), & t \in (0,1); \end{cases} \tag{6}$$

where $\mathcal{D}_0(\cdot, \cdot)$ (resp. $\mathcal{D}_1(\cdot, \cdot)$) represents the divergence measure of KL-divergence (resp. MSE); the first end-point regularization ensures the model can make correct discretizations at $t = 0$, i.e., the $1^{st}$ term in Eq. (5); and the second regularization pushes the variation of the probability space for score function to be identical to the variation of the predefined diffusion trajectories, i.e., the $2^{nd}$ term in Eq. (5). We refer the reader to Appendix A, where we theoretically show that the aforementioned requirements can be achieved by minimizing the two loss terms of $\mathcal{D}_{KL}(p(\mathbf{z}|\mathbf{x}_0), p_{\boldsymbol{\theta}}(\hat{\mathbf{z}}|\mathbf{x}_0))$ and $\|\nabla_{\mathbf{x}_t} \log p_{\boldsymbol{\theta}}(\mathbf{x}_t) - \nabla_{\mathbf{x}_t} \log p(\mathbf{x}_t)\|_2$. Given the uniqueness of discrete data (a limited number of solutions), we provide a detailed illustration of a data prediction reparameterization manner as follows.

**Data Geometry-aware Score.** Note that $\mathbf{x}_0$ belongs to a set of finite points, $\{\mathcal{P}_{\boldsymbol{\theta}}^{(c)}(\mathbf{z}), \mathbf{z} \in \{\mathbf{z}_1, \cdots, \mathbf{z}_g\}\}$. Thus, to fully capture this data geometry, we parameterize the score function into an $\mathbf{x}$-pred manner, i.e., $\nabla_{\mathbf{x}_t} \log p_{\boldsymbol{\theta}}(\mathbf{x}_t) = \frac{\mathbf{x}_t - \hat{\mathbf{x}}_{\boldsymbol{\theta}}(\mathbf{x}_t)}{\sigma_t^2}$, and further parameterize clean data estimator (i.e., $\hat{\mathbf{x}}_{\boldsymbol{\theta}}(\cdot)$) in the score function as

$$\hat{\mathbf{x}}_{\boldsymbol{\theta}}(\mathbf{x}_t) = \mathcal{P}_{\boldsymbol{\theta}}^{(c)}(\hat{\mathbf{z}}_{\boldsymbol{\theta}}), \quad \hat{\mathbf{z}}_{\boldsymbol{\theta}} = \mathcal{K}(p_{\boldsymbol{\theta}}(\hat{\mathbf{z}}|\mathbf{x}_t)), \tag{7}$$

where $\mathcal{K}(\cdot)$ indicates a discretization operator (e.g., Kronecker delta transition) to derive the exact one-hot result from $p_{\boldsymbol{\theta}}(\cdot)$. Here, we note that such a function $\mathcal{K}(\cdot)$ can be achieved by different means, e.g., sampling as a categorical distribution, resulting in a *probabilistic score*, or greedy decoding, resulting in a *deterministic score*, which is the same as traditional continuous diffusion models. Regardless of the specific method used to derive the discrete vector, the operator $\mathcal{K}(\cdot)$ is non-differentiable, rendering direct gradient descent optimization intractable. In the next section, we will introduce a relaxed evidence lower boundary (RELBO) for score learning.

---

[1]We remove the Gaussian $\epsilon$ in the modeling process because a Gaussian term is already present in the diffusion process.

**Algorithm 1:** RELBO-based score learning

| | |
|---|---|
| **Input** | : Initialized model parameter $\boldsymbol{\theta}$, dataset distribution $p(\mathbf{z}_0)$. |
| **Output** | : Updated parameter $\boldsymbol{\theta}$. |

1 **while** *not converged* **do**
2     Sampling $\mathbf{z} \sim q(\mathbf{z}), t \sim \mathcal{U}(0,1)$;
      // Token Projection & Noise Sampling
3     $\mathbf{x}_0 \leftarrow \mathcal{P}_{\boldsymbol{\theta}}^{(c)}(\mathbf{z}); \mathbf{n} \sim \mathcal{N}(0, \mathbf{I})$;
      // State Interpolation
4     $\mathbf{x}_t \leftarrow \alpha(t)\mathbf{x}_0 + \beta(t)\mathbf{n}$;
      // RELBO-based Score Optimization
5     $\mathcal{L} \leftarrow \mathcal{D}_{KL}\left(p_{\boldsymbol{\theta}}(\hat{\mathbf{z}}|\mathbf{x}_t), p(\mathbf{z}|\mathbf{x}_t)\right) + \|\mathbf{w}p_{\boldsymbol{\theta}}(\hat{\mathbf{z}}|\mathbf{x}_t) - \mathbf{w}\mathbf{z}\|_2$;
      // Gradient Descent
      $\boldsymbol{\theta} \leftarrow \boldsymbol{\theta} - \eta\nabla_{\boldsymbol{\theta}}\mathcal{L}$

**Return:** $\boldsymbol{\theta}$.

**Algorithm 2:** Language generation via data geometry-aware diffusion process

| | |
|---|---|
| **Input** | : Diffusion model $\boldsymbol{\theta}$. |
| **Output:** | Text sample $\mathbf{z}$. |

1 $\hat{\mathbf{x}}_1 \leftarrow \beta(t)\mathbf{n}, \;\; t = 1, \mathbf{n} \sim \mathcal{N}(0, \mathbf{I})$
2 **for** $t = 1 : 0$ **do**
      // $p_{\boldsymbol{\theta}}(\hat{\mathbf{z}}|\hat{\mathbf{x}}_t)$ via metric function $\mathcal{P}_{\boldsymbol{\theta}}^{(r)}(\cdot)$.
3     $\hat{\mathbf{z}}_{\boldsymbol{\theta}} \leftarrow \mathcal{K}\left(p_{\boldsymbol{\theta}}(\hat{\mathbf{z}}|\hat{\mathbf{x}}_t)\right)$
4     $\hat{\mathbf{x}}_{\boldsymbol{\theta}} \leftarrow \mathcal{P}_{\boldsymbol{\theta}}^{(c)}(\hat{\mathbf{z}}_{\boldsymbol{\theta}})$
      // Sampling $\mathbf{x}_{t-\Delta_t}$ with $2^{nd}$ order $\hat{\mathbf{x}}_{\boldsymbol{\theta}}^{(2)}$
5     $\hat{\mathbf{x}}_{t-\Delta_t} \leftarrow \alpha(t)\hat{\mathbf{x}}_{\boldsymbol{\theta}}^{(2)} + \beta(t)\left(\bar{\gamma}(t)\hat{\mathbf{n}} + \sqrt{1-\bar{\gamma}^2}\widetilde{\mathbf{n}}\right)$;
6 $\hat{\mathbf{z}} \leftarrow \mathcal{P}_{\boldsymbol{\theta}}^{(r)}(\hat{\mathbf{x}}_0)$

**Return:** $\hat{\mathbf{z}}$.

### 3.2 SCORE DIFFERENTIAL OPTIMIZATION VIA RELAXED ELBO

The training process of the diffusion model, i.e., score matching, centers on the optimization of negative log likelihood (NLL), which is bounded by the variational lower bound (VLB) (Sohl-Dickstein et al., 2015; Kingma et al., 2021). The discrete-time VLB (Ho et al., 2020) of the diffusion model is written as

$$\mathbb{E}_{p_{\boldsymbol{\theta}}}\left[-\log p_{\boldsymbol{\theta}}(\mathbf{x}_0)\right] \leq \mathbb{E}_p\left[-\sum_{t\geq 1}\log\frac{p_{\boldsymbol{\theta}}(\mathbf{x}_{t-1}|\mathbf{x}_t)}{p(\mathbf{x}_t|\mathbf{x}_{t-1})}\right] + c, \tag{8}$$

which can also be reformulated as the following continuous-time (Song et al., 2020b; 2021) form:

$$\mathbb{E}_{p_{\boldsymbol{\theta}}}\left[-\log p_{\boldsymbol{\theta}}(\mathbf{x}_0)\right] \leq \mathbb{E}_p\left[\int_0^1 g^2(t)\|s_{\boldsymbol{\theta}}(\mathbf{x}_t, t) - \nabla_{\mathbf{x}_t}\log p(\mathbf{x}_t)\|^2 dt\right] + c. \tag{9}$$

By substituting the Eqs. (7) and $\mathbb{E}_p[-\log p(\mathbf{z})] = \mathbb{E}_p[-\log p(\mathbf{z}|\mathbf{x}_0) - \log p(\mathbf{x}_0)]$ into the Eq. (9), we have

$$\text{ELBO} = \underbrace{\mathbb{E}_p\left[\int_0^1 \frac{d\lambda}{dt}\|\mathbf{w}\mathcal{K}(p_{\boldsymbol{\theta}}(\hat{\mathbf{z}}|\mathbf{x}_t)) - \mathbf{w}p(\mathbf{z}|\mathbf{x}_0)\|_2 dt\right]}_{(1^{st}) \text{ Score matching of } t\in(0,1)} + \underbrace{\mathbb{E}_p\left[\mathcal{D}_{KL}(p_{\boldsymbol{\theta}}(\hat{\mathbf{z}}|\mathbf{x}_t), p(\mathbf{z}|\mathbf{x}_t))|_{t=0}\right]}_{(2^{nd})\text{Discrete estimation at } t=0}, \tag{10}$$

where the $1^{st}$ term regularizes the variational probability flow transition; since $\mathbf{z}$ is a one-hot vector, the term $\mathbf{w}p(\mathbf{z}|\mathbf{x}_0)$ is equivalent to the expectation $\mathbb{E}_{\mathbf{z}\sim p(\mathbf{z}|\mathbf{x}_0)}[\mathbf{w}\mathbf{z}]$; and $\frac{d\lambda}{dt}$ indicates the derivative of Signal-to-Noise Ratio (SNR) with respect to time, and the $2^{nd}$ term focuses on the end-point continuous-to-discrete transition. We refer the reader to Fig. F-2 in Appendix for an intuitive illustration. To optimize the score function, we introduce a relaxed ELBO (RELBO) as

$$\text{RELBO} = \mathbb{E}_p\left\{\int_0^1 \frac{d\lambda}{dt}\left[\|\mathbf{w}\mathcal{K}(p_{\boldsymbol{\theta}}(\hat{\mathbf{z}}|\mathbf{x}_t)) - \mathbf{w}p_{\boldsymbol{\theta}}(\hat{\mathbf{z}}|\mathbf{x}_t)\|_2 + \|\mathbf{w}p_{\boldsymbol{\theta}}(\hat{\mathbf{z}}|\mathbf{x}_t) - \mathbf{w}p(\mathbf{z}|\mathbf{x}_t)\|_2\right] dt\right\} + \mathbb{E}_{p,p_{\boldsymbol{\theta}}}\mathcal{D}_{KL}, \tag{11}$$

In Appendix A.2, we theoretically show that optimizing Eq. (11) results in the following formulation of the loss function:

$$\mathcal{L} = \mathbb{E}_{(\mathbf{z},\mathbf{x})\sim p(\mathbf{Z},\mathbf{X})}\left[\mathcal{H}\left(p(\mathbf{z}|\mathbf{x}_t), p_{\boldsymbol{\theta}}(\hat{\mathbf{z}}|\mathbf{x}_t)\right) + \|\mathbf{w}p_{\boldsymbol{\theta}}(\hat{\mathbf{z}}|\mathbf{x}_t) - \mathbf{w}p(\mathbf{z}|\mathbf{x}_t)\|_2\right], \tag{12}$$

where $\mathcal{H}(\cdot, \cdot)$ refers to the cross-entropy loss, since $\mathbf{x}_t$ is interpolated between $\mathbf{x}_0$ and $\mathbf{n}$, we expect $p(\mathbf{z}|\mathbf{x}_t)$ is identical to $p(\mathbf{z}|\mathbf{x}_0)$ for $t \in [0,1)$. Algorithms 1 and 2 summarize the training and inference processes of the proposed method, respectively. Moreover, in Appendix D, we theoretically show that as the number of classes approaches infinity, optimizing Eq. (12) will make the projection of a discrete distribution approach a Gaussian.

### 3.3 MODEL SPECIFICATIONS

In the preceding sections, we have constructed a general framework for processing discrete variables via a data geometry-aware diffusion process. However, there are still several critical points to enable such a kind of model for effective learning and generation. In this section, we investigate further some design criteria for our model.

**Continuous Projector** $\mathcal{P}_c(\cdot)$. This projector can be implemented using either an embedding or a linear layer. A key challenge is that the projected states, $\mathcal{P}_c(\mathbf{z})$, are sparsely distributed and do not occupy the entire continuous space. To formalize this phenomenon, we introduce a *density* metric of discrete embeddings in a continuous space, defined as $\mathcal{F}(\mathbb{S}, \mathbb{A}) = \frac{\mu(\mathbb{S})}{\mu(\mathbb{A})}$, where $\mu(\mathbb{S})$ denotes the volume occupied by the projected states and $\mu(\mathbb{A})$ represents the total volume of the space. Assuming the space is isotropic, the total volume is proportional to $d^s$, i.e., $\mu(\mathbb{A}) \propto d^s$, where $s$ is the dimensionality of diffusion space and $d$ is the domain size in a single dimension. The volume occupied is proportional to $N_C(\delta_d)^s$, i.e., $\mu(\mathbb{S}) \propto N_C(\delta_d)^s$, where $N_C$ is the total number of classes and $\delta_d$ is the occupancy range for a single class (we refer the reader to Appendix F.1 for more discussions of $\delta_d$ and $\mu(\mathbb{S})$). Consequently, the density is given by

$$\mathcal{F}(\mathbb{S}, \mathbb{A}) \propto N_C \left( \frac{\delta_d}{d} \right)^s. \qquad (13)$$

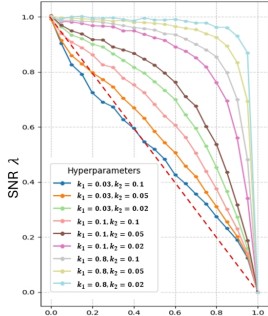

Figure 2: Illustration of projected points density on embedding continuous space with different #Dim, where we calculate the distance between neighbors. For the #Dim $> 8$, the distance between the nearest neighbors expands rapidly.

In contrast to traditional Gaussian diffusion models, which typically exhibit a density of $\mathcal{F} \approx 1$, the density of our model is sensitive to these parameters. This relationship implies that the density of our design space is directly proportional to the number of classes ($N_C$) and inversely proportional to the diffusion dimension ($s$). Note that the proposed method evolves on Euclidean continuous trajectories. Thus, the lower density results in a larger distance between any point-pairs in $\{\mathbf{x}_0\}$ and greatly amplifies the errors learned from $p_{\boldsymbol{\theta}}(\cdot)$. Due to the fact that the size of the vocabulary, i.e., $N_C$, is usually fixed, we give detailed illustrations of the effects through varying $s$. Moreover, to further regularize the distribution to be tractable and easy to analyze, *we apply an additional normalization layer that constrains the projected embeddings onto a hypersphere*.

We analyze the statistics of the average distance between nearest-neighbor pairs across varying embedding dimensions in Fig. 2. The results illustrate that high dimensionality induces large Euclidean distances and leads to asymptotic orthogonality (larger angles) among embeddings, as evidenced by the diminishing slopes for dimensions 64 and 256 compared to dimensions 4 through 8. Recall that the loss is proportional to the squared distance: $\text{ELBO} \propto \|\hat{\mathbf{x}}_0 - \mathbf{x}\|_2^2 = \|\hat{\mathbf{x}}_0\|_2^2 - 2\langle \hat{\mathbf{x}}_0, \mathbf{x} \rangle + \|\mathbf{x}\|_2^2$. Since we have constrained the embeddings to a hypersphere of radius $r$, this simplifies to:

$$\text{ELBO} \propto 2r^2(1 - \cos\varphi), \qquad (14)$$

where $r$ indicates the radius of the hypersphere, and $\varphi$ indicates the angle between $\hat{\mathbf{x}}_0$ and $\mathbf{x}$. We can conclude that higher dimensionality brings a larger distance and orthogonality between nearest neighbors and finally amplifies the geometric error of score estimation and generation.

**Customization of SDEs.** The diffusion and drift functions, i.e., $g(\cdot)$ and $f(\cdot)$, are also critical components for diffusion models. The basic design criterion for these functions is the SNR derivative $\frac{d\lambda}{dt}$. Previous research (Kingma et al., 2021; Karras et al., 2022) indicates that a diagonal SNR-t curve can benefit the model learning and data generation. Traditional Gaussian-diffusion models define the SNR as the ratio of data to noise as $\lambda(t) = \frac{\|\mathbf{x}_t\|_2}{\|\mathbf{n}\|_2}$. Due to the aforementioned spherical data

Figure 3: Illustration of the effects of different SDE configurations on the SNR-T curve, where red dashed line indicates the ideal result.

manifold design, we have $\lambda(t) = \sqrt{\frac{\|\mathbf{x}_t\|_2^2}{\|\mathbf{n}_\perp\|_2^2 + \|\mathbf{n}_{/\!/}\|_2^2}}$, where $\|\mathbf{n}_\perp\|_2$ and $\|\mathbf{n}_{/\!/}\|_2$ represent the orthogonal and parallel components of noise with respect to the direction of $\mathbf{x}_t$. Since $\mathbf{x}_0$ has been projected onto a spherical manifold, $\|\mathbf{n}_{/\!/}\|_2$ does not disturb the information in $\mathbf{x}_0$. Based on that, we utilize the angle $\varphi$ as an indicator of SNR. Then, the design of $f(\cdot)$, $g(\cdot)$, and the hypersphere is generally based on the motivation for a linear SNR-t curve (i.e., a constant $\frac{d\lambda(t)}{dt}$). Although there is indeed

an analytic solution of the distribution of $\varphi$, we apply MCMC to directly illustrate the selection of hyperparameters in Fig. 3. We configure the forward integral of diffusion SDEs as

$$\mathbf{x}_t = \alpha(t)\mathbf{x}_0 + \beta(t)\mathbf{n}, \quad \text{with} \quad \alpha(t) = k_1\sqrt{1-t} \quad \text{and} \quad \beta(t) = k_2\sqrt{t}, \tag{15}$$

where $k_1$ and $k_2$ represent the corresponding coefficients for the data and noise components; and $\mathbf{n}$ represents a standard Gaussian noise of the same size as $\mathbf{x}_0$. Due to the existence of the cross-entropy term $\mathcal{H}(p(\mathbf{z}), p_{\boldsymbol{\theta}}(\hat{\mathbf{z}}|\mathbf{x}_t))$ in the loss term as in Eq. (12), the class embedding will be uniformly distributed over the hypersphere. Thus, we randomly sample embeddings $\mathbf{x}_0$ and then apply a Gaussian perturbation to derive the intermediate variable $\mathbf{x}_t$. Finally, we calculate the angle $\varphi$ between $\mathbf{x}_0$ and $\mathbf{x}_t$ and visualize it in Fig. 3, accompanied by a red dashed line as the ideal result. The results indicate that $\lambda$ is quite sensitive to the hyperparameters. To derive a linear $\lambda - t$ curve to get rid of $\frac{d\lambda}{dt}$, we select $k_1 = 3e^{-2}$ and $k_2 = 1e^{-1}$, which is also the closest to the ideal curve.

**Reverse Integration**. To simplify the integration of SDE-based diffusion formulation, we directly design the reverse integral form of $\mathbf{x}_t = F(t)\mathbf{x}_0 + G(t)\mathbf{n}$, with corresponding coefficients of $F(t)$ and $G(t)$. Based on Appendix B, we can further reformulated the reverse integral form as

$$\mathbf{x}_t = \alpha(t)\hat{\mathbf{x}}_{\boldsymbol{\theta}}(\mathbf{x}_{t+\Delta t}) + \beta(t)\left(\bar{\gamma}(t)\hat{\mathbf{n}}(\mathbf{x}_{t+\Delta t}, \hat{\mathbf{x}}_{\boldsymbol{\theta}}) + \sqrt{1 - \bar{\gamma}(t)}\tilde{\mathbf{n}}\right), \tag{16}$$

where $\alpha(t)$ and $\beta(t)$ represent the coefficients for the data and noise components as defined in Eq. (15); $\bar{\gamma}(t)$ refers to a coefficient to control the ratio between the stochastic and deterministic components; $\hat{\mathbf{n}}(\mathbf{x}_{t+\Delta t}, \hat{\mathbf{x}}_{\boldsymbol{\theta}}) = (\mathbf{x}_{t+\Delta t} - \alpha(t)\hat{\mathbf{x}}_{\boldsymbol{\theta}})/\beta(t)$ is the estimated Gaussian noise derived from $\mathbf{x}_{t+\Delta t}$ and $\hat{\mathbf{x}}_{\boldsymbol{\theta}}$; and $\tilde{\mathbf{n}}$ is a newly sampled Gaussian noise. Note that this reverse integral formulation has the same probability flow as the *general $\gamma$-SDE* $d\mathbf{x}_t = \left[f(\mathbf{x}_t, t) - \frac{1+\gamma}{2}g^2(t)\nabla_{\mathbf{x}_t}\log p(\mathbf{x}_t)\right]dt + \gamma g(t)d\mathcal{B}_t$. During the implementation, we also integrate it with a second-order Heun integrator method (Lu et al., 2022) that averages the local scores to boost generation accuracy.

# 4 EXPERIMENTS

**Datasets.** Two substantial and distinct datasets, i.e., One Billion Word Benchmark (LM1B) and OpenWebText, were utilized to comprehensively evaluate the performance of our models. Specifically, the LM1B dataset (Chelba et al., 2013) is a large-scale corpus containing approximately one billion words, derived from the WMT 2011 News Crawl dataset. The OpenWebText dataset (Gokaslan et al., 2019) offers a more contemporary and diverse collection of text. It is an open-source replication of the WebText corpus, which was used to train the GPT-2 model. The corpus was generated by scraping web pages linked from Reddit submissions that received a minimum of three upvotes, resulting in a dataset that reflects a wide array of topics and writing styles found on the internet. The dataset contains approximately 38 GB of text.

**Training Details.** We adopted the same training settings as in previous work (Jo & Hwang, 2025; Sahoo et al., 2024; Lou et al., 2023). For the LM1B dataset, we used the same tokenizer as (He et al., 2022; Jo & Hwang, 2025), with a context size of 128 and a vocabulary size of around 30,000. The network backbone is a diffusion transformer architecture (Peebles & Xie, 2023) with rotary positional embeddings (Su et al., 2023). Moreover, for the OpenWebText dataset, we standardized the sequence length of text to 1024 and used the GPT2 tokenizer (Radford et al., 2019) with a vocabulary size around 50000. We trained our models with a batch size of 512 (resp. 512) with a learning rate of 3e-4, 2500 warming up steps for 300K (resp. 1M) steps on the LM1B (resp. OpenWebText) dataset with 12×RTX 4090 (resp. 8×RTX Pro 6000) GPUs, respectively.

**Ranking Metrics.** The accurate evaluation of discrete and continuous language models requires a fair and reliable metric. In Appendix C, we give the detailed formulation for likelihood calculation, which is based on (Song et al., 2020b). Moreover, inspired by recent advances in LLMs, we propose several metrics to evaluate the quality of generated samples. Specifically, Perceptual Score uses an LLM (Gemini-2.5-pro (Comanici et al., 2025)) as a proxy for human judgment (Prompt is shown in Appendix G.1). The methodology involves ranking the outputs of various models against a designated baseline model. The final score is then computed by averaging these relative rankings. To ensure impartiality and mitigate bias, all samples are evaluated anonymously and presented in a randomized order. We refer the reader to Appendix G for a detailed description of the implementation. Furthermore, we also compare the PPL of the generated samples using different open-source pretrained LLMs, including Qwen-3 (Qwen3-30B-A3B-Instruct-2507) (Yang et al., 2025) and DeepSeek-R1 (DeepSeek-R1-Distill-Llama-8B) (Guo et al., 2025).

Table 3: Quantitative comparison of different language models on the OpenWebText dataset. The pretrained models AR and MDLM are from (Sahoo et al., 2024), where "$-1k$" (resp."$-5k$") denotes the results with the number of diffusion inference steps, also called the number of function evaluation (NFE) in diffusion-related papers, of "$-1k$" (resp."$-5k$"). † indicates the method with the Heun predictor-corrector integrator. The $\bar{\gamma}_1$ to $\bar{\gamma}_4$ are 0.99, 0.98, 0.998, and 0.9, respectively. "PPL-L" denotes the perplexity calculated by trained models on testing data. Note that the likelihood calculation of the proposed method is generally based on the integral of the density transition over diffusion ODE trajectories, which is different from discrete methods. The metrics of "Rank", "PPL-Q", and "PPL-D" measure the generation results by the LLMs of Gemini-2.5-pro, Qwen3-30B, and DeepSeek-R1, respectively.

| Methods | # Param | Train Iter | Rank ↓ | PPL-L ↓ | PPL-Q ↓ | PPL-D ↓ |
|---|---|---|---|---|---|---|
| **AR** | 110M | 1.0M | 0.50 | 17.56 | 58.93 | 87.59 |
| **SEDD** | 110M | 1.0M | 1.0 | 24.56 | 76.66 | 123.21 |
| **MDLM-1K** | 110M | 1.0M | 0.62 | 23.83 | 69.60 | 98.27 |
| **MDLM-5K** | 110M | 1.0M | 0.54 | | 39.51 | 51.49 |
| **EDLM** | 110M | 1.0M | – | 21.52 | – | – |
| **LanGeo-1k-$\bar{\gamma}_1$** † | 110M | 1.0M | 0.65 | | 27.72 | 39.94 |
| **LanGeo-1k-$\bar{\gamma}_2$** † | 110M | 1.0M | 0.59 | 17.1 | 36.53 | 70.21 |
| **LanGeo-5k-$\bar{\gamma}_3$** † | 110M | 1.0M | 0.63 | | 18.24 | 25.92 |
| **LanGeo-5k-$\bar{\gamma}_4$** | 110M | 1.0M | 0.40 | | 32.16 | 45.86 |

**Baseline Methods.** We compare our method with the latest autoregressive and diffusion models, including SEDD (Lou et al., 2023), MDLM (Sahoo et al., 2024), EDLM (Xu et al., 2024), which are discrete, and RDLM (Jo & Hwang, 2025), which is continuous. We exclude Plaid (Gulrajani & Hashimoto, 2023) from comparison as it was trained on significantly shorter sequences (256 tokens) than current state-of-the-art standards. We also compare with the Transformer-based AR model (Vaswani et al., 2017).

## 4.1 EXPERIMENTAL COMPARISON ON LM1B

Experimental results are shown in Table 2. The likelihood-based PPL illustrated as "PPL-L" indicates the superior performance of the proposed method. Due to the fact that most pre-trained models on LM1B are unavailable and the size of this benchmark is small, we primarily compare our method with RDLM and the pretrained GPT-2 regarding the proposed Rank score, primarily for toy experimental validation. RDLM's average

Table 2: Quantitative comparisons of different language models on the LM1B dataset. "GPT2" is trained on the WebText dataset.

| Methods | Charac | # Param | Rank ↓ | PPL-L↓ |
|---|---|---|---|---|
| **GPT2-L** | **AR** | 110M | 0.0 | – |
| Transformer | | 110M | – | 22.32 |
| **SEDD** | **Discrete Diff** | 110M | – | 32.79 |
| **MDLM** | | 110M | – | 27.04 |
| **Diffusion-LM** | **Continuous Diff** | 110M | – | 118.62 |
| **RDLM** | | 110M | 1.00 | 29.72 |
| **LanGeo (Ours)** | | 110M | 0.51 | 21.85 |

rank of 1.00 indicates that its generated samples were consistently ranked lower than GPT-2 and our method. Experimental results indicate that our method achieves SOTA performance on the continuous domain and that our generated samples outperform all RDLM's results with a 100% win rate. Although our method does not surpass even GPT-2, we want to note that GPT-2 is trained on a larger dataset with many more iterations. Here, we list it to give more comparisons for comprehensive evaluations.

## 4.2 EXPERIMENTAL COMPARISON ON OPENWEBTEXT

We further validate the performance of our method via a large-scale OpenWebText dataset. As shown in Table 3, our method **significantly outperforms** all compared methods, even the autoregressive baseline, **regarding all metrics**. Specifically, it achieves comparable results with MDLM, SEDD, and even outperforms part of the samples generated from autoregressive transformers[2], which indicates its strong potential. The divergence between the Rank and PPL also indicates the necessity of introducing other metrics to fairly and accurately measure the generation results. Moreover, we refer the reader to Appendix J for the generation results.

---

[2]For the model of the autoregressive and SEDD, we utilize the implementation from MDLM (Sahoo et al., 2024).

We also want to note that our method is a plain continuous language model and can be integrated with modern training techniques, such as consistency training (Song et al., 2023) or discriminative score distillation (Sauer et al., 2024), for further performance enhancement.

## 4.3 Ablation Studies

Although most parts of the proposed framework and hyperparameters, e.g., $\alpha(t)$ and $\beta(t)$, are designed in a theoretically grounded manner, there are still some parts that need to be validated.

**NFE and stochastic factor** $\bar{\gamma}(t)$ are critical hyperparameters to balance the generation quality and speed. We carry out experiments to validate their effectiveness. Experimental results are shown in Table 4. Results illustrate that the large computational resources provided by the additional inference steps can significantly enhance the diffusion performance. Moreover, a larger number of diffusion steps should adapt with a smaller stochastic factor $\bar{\gamma}(t)$ to achieve better performance. It's also theoretically reasonable that the intensity of the stochastic component should change adaptively with the step size. Moreover, we can also compare the proposed method with SOTA methods in Table 3. Regarding the metric of generation PPL, our method, with only **100** steps, can outperform the baseline MDLM-1K with **10×** acceleration. This can be attributed to our continuous diffusion framework design, which enables powerful integrators and can be further enhanced with parallel solvers (Shih et al., 2023; Lu et al., 2025).

Table 4: Ablation studies of NFE and $\bar{\gamma}(t)$ on the OpenWebText dataset. All with Heun integrator.

| NFE | $\bar{\gamma}(t)$ | Rank ↓ | PPL-Q↓ | PPL-D↓ |
|---|---|---|---|---|
| 100 | 0.8 | 0.70 | 69.84 | 101.82 |
| 100 | 0.9 | 0.72 | 52.37 | 76.26 |
| 200 | 0.9 | 0.70 | 54.18 | 78.71 |
| 200 | 0.95 | 0.62 | 39.41 | 56.97 |
| 500 | 0.95 | 0.76 | 47.80 | 70.21 |
| 500 | 0.98 | 0.66 | 31.02 | 45.26 |
| 1000 | 0.98 | 0.65 | 36.53 | 70.21 |
| 1000 | 0.99 | 0.59 | 27.72 | 45.26 |

**The proposed rank metric** is also a contribution of this paper. To begin with, the inconsistency between the rank and PPL as shown in Tables 3 and 4 shows the necessity of measuring the generation quality. Specifically, LanGeo-5k-$\bar{\gamma}_4$ has similar PPL with LanGeo-5k-$\bar{\gamma}_3$. However, its rank is significantly lower than 5k-$\bar{\gamma}_3$ and 1k-$\bar{\gamma}_2$. Such a phenomenon suggests that the PPL is incomplete in measuring the quality of generation. To further demonstrate the necessity of the proposed rank metric, we append further non-cherry-picked results (the first generation result without any selection) in Appendix J.

## 5 Related Work

**Language Model.** The dominant paradigm in generative natural language processing is the autoregressive model, epitomized by the Transformer architecture. Introduced by (Vaswani et al., 2017), the Transformer's self-attention mechanism became the cornerstone for a new generation of models capable of capturing long-range dependencies in text. Models such as the Generative Pre-trained Transformer (GPT) series (Radford et al., 2018; Brown et al., 2020) have demonstrated unparalleled performance by pre-training on vast web-scale corpora and then fine-tuning for specific tasks. These models generate text in a strictly sequential, left-to-right manner, factorizing the joint probability of a sequence into a product of conditional probabilities: $\mathbf{P}(x) = \Pi_{i=1}^{T} P(x_i|x_{<i})$. While immensely successful, this sequential dependency imposes fundamental limitations, including error propagation and an inability to revise or refine previously generated tokens, motivating the exploration of alternative, more holistic generation frameworks.

**Diffusion Models in Continuous Domains.** Diffusion models (Sohl-Dickstein et al., 2015) have recently emerged as a remarkably powerful class of generative models, particularly in continuous domains such as image and audio synthesis. Their modern resurgence was catalyzed by (Sohl-Dickstein et al., 2015) with Denoising Diffusion Probabilistic Models (DDPMs), which demonstrates state-of-the-art image generation quality. The core principle involves a fixed forward process that systematically corrupts data with Gaussian noise over a series of timesteps and a learned reverse process that iteratively denoises a random noise vector back into a coherent data sample. Subsequent work has focused on accelerating the slow, iterative sampling process (Song et al., 2020a) and improving computational efficiency by performing the diffusion process in a compressed latent space rather than the high-dimensional pixel space (Rombach et al., 2022). The success of these models in generating complex, high-fidelity data structures through a gradual refinement process underscores their potential for tasks that require global coherence.

**Diffusion Models for Natural Language.** Adapting diffusion models to the discrete and symbolic nature of text presents a significant challenge. Early and influential work in this area, Diffusion-LM (Li et al., 2022), pioneered the application of diffusion directly in the continuous word embedding space. This approach performs the forward noising and reverse denoising process on sequences of word vectors, followed by a rounding step to map the final continuous representations back to discrete vocabulary tokens.

Following this, several alternative frameworks have been proposed. Some models, such as LD4PG (Zou et al., 2024), introduce a variational autoencoder (VAE) (Kingma & Welling, 2013) to learn a structured continuous latent space for entire sentences, where the diffusion process subsequently takes place. This offers a more compressed and potentially more semantically meaningful space for generation. Other approaches have explored discrete diffusion models (Austin et al., 2021) that define the noising process directly over the discrete token space, bypassing the need for a continuous embedding space entirely. Despite these innovations, a unifying challenge remains the significant inference latency due to the iterative sampling process, which is an active area of research. Current work continues to explore architectural improvements and more efficient sampling strategies to close the performance gap with highly optimized auto-regressive models.

## 6 DISCUSSION

**A Plain Model**. Note that the proposed method is a plain continuous diffusion model for language generation. The competitive performance indicates the potential of continuous modeling for discrete data. Moreover, based on the continuous modeling, we can adapt the mature image diffusion techniques directly to the language generation field.

**The Scaling Law**. Validation of scaling performance for a new generation scheme is quite challenging. However, we hypothesize that our model will scale positively, consistent with trends observed in other large-scale generative models, particularly within the diffusion model landscape. Previous work has shown that scaling up generally leads to significant gains in output quality, coherence, and fidelity. Given our model's architectural similarities to these proven frameworks, we expect larger versions would produce more fluent, contextually relevant, and nuanced text. Therefore, we identify the empirical validation of these scaling laws as a critical direction for future research.

**Convergence of the Geometry-aware Score to the Gaussian Score as Class Count Increases.** In Appendix D, we theoretically prove that as the number of classes approaches $\infty$, our geometry-aware score matching is also approaching a Gaussian score. This also motivates us to rethink the proposed method as a bridge to connect the continuous appearance of the classic continuous diffusion model. Due to the explicit modeling of the data manifold, applying our geometry-aware score matching may also aid in modeling continuous data.

**Integrator.** Although the experimental results with integrator, e.g., Heun predictor-corrector have higher PPL performance, their perceptual quality is quite low as measured by the rank metric. Thus, there is also necessity to design further advanced integrators for our dat geometry-aware score matching.

## 7 CONCLUSION

We have presented a novel, concise and effective approach to achieve discrete language generation via data geometry-aware score-matching. This method establishes a crucial link between the discrete diffusion process and the well-established Euclidean Gaussian diffusion framework. Score optimization is made tractable by utilizing a relaxed evidence lower bound (RELBO). For evaluation, we addressed existing challenges by introducing a rank-based metric, which employs online large language models for a more robust assessment of sample quality. The proposed method offers several advantages, e.g., competitive generation performance, and a smaller number of embedding parameters (only 0.78%). Moreover, a core principle of our design is conciseness; the framework has been simplified by systematically reducing unnecessary hyperparameters and modules. It is our hope that this contribution will encourage broader exploration of continuous generative models within discrete data domains.

**Ethics statement.** The authors hereby confirm that we have thoroughly read and understood Code of Ethics of ICLR 2026. We acknowledge the importance of its general principles, including the commitment to Responsible Stewardship, upholding high standards of scientific excellence, contributing to societal well-being, avoiding harm, and maintaining honesty, fairness, and transparency. We agree to abide by this code in all our contributions, submissions, and interactions within the ICLR community, ensuring our work aligns with these ethical guidelines for responsible research.

**Reproducibility statement.** We have illustrated the critical components in detail and selection of hyper-parameters in Sec. 3.3. Moreover, we have attached the implementation of proposed Rank metric.

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

CONTENTS

# A CONTINUOUS-TIME PROBABILITY FLOW DERIVATION

## A.1 PROBABILITY FLOW

We start from the following regularization

$$
\mathcal{L}_t = \left\{
\begin{array}{ll}
\mathcal{D}\left(p_{\boldsymbol{\theta}}(\hat{\mathbf{z}}|\mathbf{x}_t), p(\mathbf{z}|\mathbf{x}_t)\right), & t = 0; \\
\mathcal{D}\left(\frac{dp_{\boldsymbol{\theta}}(\mathbf{x}_t)}{dt}, -\frac{d(f(\mathbf{x}_t,t)p(\mathbf{x}_t))}{d\mathbf{x}_t} + \frac{1}{2}\frac{d^2(g^2(\mathbf{x}_t,t)p(\mathbf{x}_t))}{d\mathbf{x}_t^2}\right), & t \in (0,1),
\end{array}
\right.
\tag{17}
$$

With respect to the Fokker–Planck equation (Risken, 1989), we have

$$
\frac{dp_{\boldsymbol{\theta}}(\mathbf{x}_t)}{dt} = -\frac{d(f(\mathbf{x}_t,t)p_{\boldsymbol{\theta}}(\mathbf{x}_t))}{d\mathbf{x}_t} + \frac{1}{2}\frac{d^2(g^2(\mathbf{x}_t,t)p(\mathbf{x}_t))}{d\mathbf{x}_t^2}.
\tag{18}
$$

Thus, the regularization becomes

$$
\mathcal{L}_t = \left\{
\begin{array}{ll}
\mathcal{D}_0\left(p_{\boldsymbol{\theta}}(\hat{\mathbf{z}}|\mathbf{x}_t), p(\mathbf{z}|\mathbf{x}_t)\right), & t = 0; \\
\mathcal{D}_1\left(-\frac{d(f(\mathbf{x}_t,t)p_{\boldsymbol{\theta}}(\mathbf{x}_t))}{d\mathbf{x}_t} + \frac{1}{2}\frac{d^2(g^2(\mathbf{x}_t,t)p_{\boldsymbol{\theta}}(\mathbf{x}_t))}{d\mathbf{x}_t^2}, -\frac{d(f(\mathbf{x}_t,t)p(\mathbf{x}_t))}{d\mathbf{x}_t} + \frac{1}{2}\frac{d^2(g^2(\mathbf{x}_t,t)p(\mathbf{x}_t))}{d\mathbf{x}_t^2}\right), & t \in (0,1),
\end{array}
\right.
\tag{19}
$$

**Terminal Condition** ($t = 0$). The objective at $t = 0$ is to minimize:

$$
\mathcal{L}_0 = \mathcal{D}\left(p_{\boldsymbol{\theta}}(\mathbf{x}_0), p(\mathbf{z})\right)
\tag{20}
$$

Minimizing any divergence $\mathcal{D}(P, Q)$ to its lower bound of 0 requires that its arguments be identical.

$$
\min_{\boldsymbol{\theta}} \mathcal{L}_0 \implies \mathcal{L}_0 = 0 \iff p_{\boldsymbol{\theta}}(\mathbf{x}_0) = p(\mathbf{z})
\tag{21}
$$

This is the same condition required to minimize the target KL-divergence:

$$
\min_{\boldsymbol{\theta}} \mathcal{D}_{KL}((p_{\boldsymbol{\theta}}(\hat{\mathbf{z}}|\mathbf{x}_t), p(\mathbf{z}|\mathbf{x}_t)) \implies \mathcal{D}_{KL} = 0 \iff p_{\boldsymbol{\theta}}(\hat{\mathbf{z}}|\mathbf{x}_t) = p(\mathbf{z}|\mathbf{x}_t)
\tag{22}
$$

Thus, minimizing $\mathcal{L}_0$ is equivalent to regularizing $\mathcal{D}_{KL}(p(\mathbf{z}), p_{\boldsymbol{\theta}}(\mathbf{x}_0))$.

**Path Regularization** ($t \in (0, 1)$). The path objective minimizes:

$$
\mathcal{L}_t = \mathcal{D}\left(-\frac{d(f(\mathbf{x}_t,t)p_{\boldsymbol{\theta}}(\mathbf{x}_t))}{d\mathbf{x}_t} + \frac{1}{2}\frac{d^2(g^2(\mathbf{x}_t,t)p_{\boldsymbol{\theta}}(\mathbf{x}_t))}{d\mathbf{x}_t^2}, -\frac{d(f(\mathbf{x}_t,t)p(\mathbf{x}_t))}{d\mathbf{x}_t} + \frac{1}{2}\frac{d^2(g^2(\mathbf{x}_t,t)p(\mathbf{x}_t))}{d\mathbf{x}_t^2}\right),
\tag{23}
$$

Considering it from the reverse trajectory that $p(\mathbf{x}_1) = p_{\boldsymbol{\theta}}(\mathbf{x}_1) = \mathcal{N}$. Moreover, according to the reverse Fokker–Planck equation of

$$
\frac{dp}{d\hat{t}} = -\frac{d}{dx}\left[f(x,t)p(x) - \left(\frac{p(x)dg^2(t)}{dx} + \frac{g^2(t)dp(x)}{dx}\right)\right] - \frac{1}{2}\frac{d^2\left[g^2(t)p(x)\right]}{dx^2}.
\tag{24}
$$

It corresponds to the reverse SDEs (Song et al., 2020b) of

$$
d\mathbf{x} = \left[f(\mathbf{x},t) - g^2(\mathbf{x},t)\nabla_x \log p(\mathbf{x})\right]dt + g(\mathbf{x},t)d\widetilde{\mathcal{B}}_t,
\tag{25}
$$

Therefore, the target norm is minimized to zero.

$$
\|\nabla_{\mathbf{x}_t} \log p_{\boldsymbol{\theta}}(\mathbf{x}_t) - \nabla_{\mathbf{x}_t} \log p(\mathbf{x}_t)\|_2 = 0
\tag{26}
$$

## A.2 SCORE FUNCTION OPTIMIZATION VIA RELBO

We first give the detailed derivation of RELBO from ELBO

$$\text{ELBO} = \mathbb{E}_p \left[ \int_0^1 \frac{d\lambda}{dt} \left\| \mathbf{w}\mathcal{K}(p_{\boldsymbol{\theta}}(\hat{\mathbf{z}}|\mathbf{x}_t)) - \mathbf{w}p(\mathbf{z}|\mathbf{x}_0) \right\|_2 dt \right] + \mathbb{E}_{p,p_\theta} \mathcal{D}_{KL}(p_{\boldsymbol{\theta}}(\hat{\mathbf{z}}|\mathbf{x}_t), p(\mathbf{z}|\mathbf{x}_t))|_{t=0}, \tag{27}$$

$$= \mathbb{E}_p \left[ \int_0^1 \frac{d\lambda}{dt} \left\| \mathbf{w}\mathcal{K}(p_{\boldsymbol{\theta}}(\hat{\mathbf{z}}|\mathbf{x}_t)) - \mathbf{w}p_{\boldsymbol{\theta}}(\hat{\mathbf{z}}|\mathbf{x}_t) + \mathbf{w}p_{\boldsymbol{\theta}}(\hat{\mathbf{z}}|\mathbf{x}_t) - \mathbf{w}p(\mathbf{z}|\mathbf{x}_0) \right\|_2 dt \right]$$
$$+ \mathbb{E}_{p,p_\theta} \mathcal{D}_{KL}(p_{\boldsymbol{\theta}}(\hat{\mathbf{z}}|\mathbf{x}_t), p(\mathbf{z}|\mathbf{x}_t))|_{t=0} \tag{28}$$

$$\leq \mathbb{E}_p \left\{ \int_0^1 \frac{d\lambda}{dt} \left[ \left\| \mathbf{w}\mathcal{K}(p_{\boldsymbol{\theta}}(\hat{\mathbf{z}}|\mathbf{x}_t)) - \mathbf{w}p_{\boldsymbol{\theta}}(\hat{\mathbf{z}}|\mathbf{x}_t) \right\|_2 + \left\| \mathbf{w}p_{\boldsymbol{\theta}}(\hat{\mathbf{z}}|\mathbf{x}_t) - \mathbf{w}p(\mathbf{z}|\mathbf{x}_0) \right\|_2 \right] dt \right\}$$
$$+ \mathbb{E}_{p,p_\theta} \mathcal{D}_{KL}(p_{\boldsymbol{\theta}}(\hat{\mathbf{z}}|\mathbf{x}_t), p(\mathbf{z}|\mathbf{x}_t))|_{t=0} \tag{29}$$

$$= \text{RELBO} \tag{30}$$

Then, we further demonstrate the correlation between the proposed RELBO and the loss function. We start with RELBO shown as below

$$\text{RELBO} = \mathbb{E}_p \int_0^1 \frac{d\lambda}{dt} \left[ \underbrace{\left\| \mathbf{w}\mathcal{K}(p_{\boldsymbol{\theta}}(\hat{\mathbf{z}}|\mathbf{x}_t)) - \mathbf{w}p_{\boldsymbol{\theta}}(\hat{\mathbf{z}}|\mathbf{x}_t) \right\|_2}_{(A)} + \underbrace{\left\| \mathbf{w}p_{\boldsymbol{\theta}}(\hat{\mathbf{z}}|\mathbf{x}_t) - \mathbf{w}\mathbf{z} \right\|_2}_{(B)} \right] dt + \underbrace{\mathbb{E}_{p,p_\theta} \mathcal{D}_{KL}}_{(C)}, \tag{31}$$

Due to the Term (C) can be directly achieved by regularizing the cross-entropy, we focus on the first two terms as

$$\mathbb{E}_p \int_0^1 \frac{d\lambda}{dt} \left[ \left\| \mathbf{w}\mathcal{K}(p_{\boldsymbol{\theta}}(\hat{\mathbf{z}}|\mathbf{x}_t)) - \mathbf{w}p_{\boldsymbol{\theta}}(\hat{\mathbf{z}}|\mathbf{x}_t) \right\|_2 + \left\| \mathbf{w}p_{\boldsymbol{\theta}}(\hat{\mathbf{z}}|\mathbf{x}_t) - \mathbf{w}p(\mathbf{z}|\mathbf{x}_t) \right\|_2 \right] dt \tag{32}$$

$$\simeq \mathbb{E}_{(\mathbf{x}_t,\mathbf{z}) \sim p(\mathbf{x}_t,\mathbf{z})} \frac{d\lambda}{dt} \left( \mathcal{H}(p_{\boldsymbol{\theta}}(\hat{\mathbf{z}}|\mathbf{x}_t)) + \left\| \mathbf{w}p_{\boldsymbol{\theta}}(\hat{\mathbf{z}}|\mathbf{x}_t) - \mathbf{w}p(\mathbf{z}|\mathbf{x}_t) \right\|_2 \right) dt.$$

Here, we made such approximation due to that minimizing $\left\| \mathbf{w}\mathcal{K}(p_{\boldsymbol{\theta}}(\hat{\mathbf{z}}|\mathbf{x}_t)) - \mathbf{w}p_{\boldsymbol{\theta}}(\hat{\mathbf{z}}|\mathbf{x}_t) \right\|_2$ can be achieved via minimize the entropy of $p_{\boldsymbol{\theta}}(\hat{\mathbf{z}}|\mathbf{x}_t)$, i.e., $\mathcal{H}(p_{\boldsymbol{\theta}}(\hat{\mathbf{z}}|\mathbf{x}_t))$. Furthermore, to minimize $\mathcal{H}(p_{\boldsymbol{\theta}}(\hat{\mathbf{z}}|\mathbf{x}_t))$, we propose to regularize the divergence between $p_{\boldsymbol{\theta}}(\hat{\mathbf{z}}|\mathbf{x}_t)$ and a low entropy target, i.e., $p(\mathbf{z}|\mathbf{x_0})$ ($\mathcal{H}(p(\mathbf{z}|\mathbf{x_0})) = 0$). Then, through merging (A) and (C) terms together, we can derivative the following loss

$$\mathcal{L} = \mathbb{E}_{(\mathbf{z},\mathbf{x}) \sim p(\mathbf{z},\mathbf{x})} \left[ \mathcal{H}\left(p_{\boldsymbol{\theta}}(\hat{\mathbf{z}}|\mathbf{x}_t), p(\mathbf{z}|\mathbf{x}_t)\right) + \left\| \mathbf{w}p_{\boldsymbol{\theta}}(\hat{\mathbf{z}}|\mathbf{x}_t) - \mathbf{w}p(\mathbf{z}|\mathbf{x}_t) \right\|_2 \right], \tag{33}$$

## B  REVERSE INTEGRAL FORM DERIVATION

A forward diffusion process is described by the Stochastic Differential Equation (SDE):

$$d\mathbf{x}_t = f(\mathbf{x}_t, t)dt + g(t)d\mathcal{B}_t, \quad t \in [0, 1] \tag{34}$$

where $f(\mathbf{x}_t, t)$ is the drift, $g(t)$ is the diffusion coefficient, and $d\mathcal{B}_t$ is a standard Wiener process.

The corresponding reverse-time SDE is given by:

$$d\mathbf{x}_t = [f(\mathbf{x}_t, t) - g^2(t)\nabla_{\mathbf{x}_t} \log p_t(\mathbf{x}_t)]dt + g(t)d\widetilde{\mathcal{B}}_t \tag{35}$$

where $\nabla_{\mathbf{x}_t} \log p_t(\mathbf{x}_t)$ is the score function of the marginal density $p_t(\mathbf{x}_t)$.

The evolution of the probability density $p(\mathbf{x}, t)$ is governed by the Fokker-Planck equation:

$$\frac{\partial p(\mathbf{x}, t)}{\partial t} = -\nabla_{\mathbf{x}} \cdot [f(\mathbf{x}, t)p(\mathbf{x}, t)] + \frac{1}{2}g^2(t)\nabla_{\mathbf{x}}^2 p(\mathbf{x}, t) \tag{36}$$

The reverse drift term is derived by ensuring the Fokker-Planck equation for the time-reversed process is consistent, leading to the drift component as $f(\mathbf{x}_t, t) - g(t)^2\nabla_{\mathbf{x}_t} \log p(\mathbf{x}_t)$.

### B.1 ANALYSIS OF THE GENERAL $\gamma$-MODULATED SDE

A generalized reverse SDE, termed Modulated-SDE (M-SDE), can be formulated as:

$$d^{(\gamma)}\mathbf{x}_t = \left[ f(\mathbf{x}_t, t) - \frac{1+\gamma^2}{2} g^2(t)\nabla_{\mathbf{x}_t} \log p_t(\mathbf{x}_t) \right] dt + \gamma g(t) d\widetilde{\mathcal{B}}_t \tag{37}$$

where $\gamma \geq 0$ controls the stochasticity. The Fokker-Planck equation for this reverse process is:

$$\frac{\partial p(\mathbf{x}_t)}{\partial t} = \nabla_{\mathbf{x}_t} \cdot [f(\mathbf{x}_t, t)p(\mathbf{x}_t)] - \frac{1+\gamma^2}{2}\nabla_{\mathbf{x}_t} \cdot [g^2(t)\nabla p] + \frac{\gamma^2}{2}\nabla_{\mathbf{x}_t}^2 [g^2(t)p(\mathbf{x}_t)], \tag{38}$$

which can be simplified to

$$\frac{\partial p(\mathbf{x}_t)}{\partial t} = \nabla_{\mathbf{x}_t} \cdot [f(\mathbf{x}_t, t)p(\mathbf{x}_t)] - \frac{1}{2}g^2(t)\nabla_{\mathbf{x}_t}^2 p(\mathbf{x}_t). \tag{39}$$

This final form is independent of $\gamma$, proving that all SDEs in this family share the same probability flow.

### B.2 THE DISCRETIZED INTEGRAL SOLVER

For a process with linear drift $f(\mathbf{x}_t, t) = \frac{\dot{F}(t)}{F(t)}\mathbf{x}_t$, we use a semi-linear solver with the surrogate function $\mathcal{F}(\mathbf{x}_t, t) = \frac{\mathbf{x}_t}{F(t)}$. Applying Itô's lemma to the $\gamma$-SDE yields:

$$d\mathcal{F}(\mathbf{x}_t, t) = -\frac{1}{F(t)}\frac{1+\gamma^2}{2}g^2(t)\nabla_{\mathbf{x}_t} \log p(\mathbf{x}_t)dt + \frac{\gamma g(t)}{F(t)}d\widetilde{\mathcal{B}}_t \tag{40}$$

Integrating from $t + \Delta t$ to $t$ and applying a first-order approximation gives the general integral solver:

$$\mathbf{x}_t \approx \frac{F(t)}{F(t+\Delta t)}\mathbf{x}_{t+\Delta t} + \frac{F(t)}{F(t+\Delta t)}\frac{1+\gamma^2}{2}g^2(t+\Delta t)\nabla_{\mathbf{x}} \log p(\mathbf{x}_{t+\Delta t})\Delta t - \frac{F(t)}{F(t+\Delta t)}\gamma g(t+\Delta t)\sqrt{\Delta t}\mathbf{n} \tag{41}$$

### B.3 CHARACTERIZING THE LINEAR FORWARD PROCESS

The specified forward process is:

$$\mathbf{x}_t = F(t)\mathbf{x}_0 + G(t)\mathbf{N}, \quad \mathbf{N} \sim \mathcal{N}(0, \mathbf{I}) \tag{42}$$

The corresponding SDE has drift and diffusion coefficients:

$$f(\mathbf{x}_t, t) = \frac{\dot{F}(t)}{F(t)}\mathbf{x}_t \tag{43}$$

$$g^2(t) = 2G(t)\dot{G}(t) - 2\frac{\dot{F}(t)}{F(t)}G(t)^2 \tag{44}$$

The score function of the transition kernel $p(\mathbf{x}_t|\mathbf{x}_0) = \mathcal{N}\left(\mathbf{x}_t; F(t)\mathbf{x}_0, G(t)^2\mathbf{I}\right)$ is:

$$\nabla_{\mathbf{x}_t} \log p(\mathbf{x}_t|\mathbf{x}_0) = -\frac{\mathbf{x}_t - F(t)\mathbf{x}_0}{G(t)^2} \tag{45}$$

This allows the score to be approximated using a neural network $\hat{\mathbf{x}}_{\boldsymbol{\theta}}(\mathbf{x}_t, t)$ trained to predict $\mathbf{x}_0$, as the following learned score

$$\nabla_{\mathbf{x}_t} \log p_{\boldsymbol{\theta}}(\mathbf{x}_t) \leftarrow -\frac{\mathbf{x}_t - F(t)\hat{\mathbf{x}}_{\boldsymbol{\theta}}(\mathbf{x}_t, t)}{G(t)^2}. \tag{46}$$

The goal is to express the reverse integral form purely in terms of the predicted clean data $\hat{\mathbf{x}}_{\boldsymbol{\theta}}$ and noise terms, eliminating the explicit dependency on the previous state $\mathbf{x}_{t+\Delta t}$.

### B.4 REPARAMETERIZATION OF THE PREVIOUS STATE

From the forward process at time $t+\Delta t$, we can estimate the noise component, $\hat{n}$, using the model's prediction of the clean data, $\hat{\mathbf{x}}_0 = \hat{\mathbf{x}}_{\boldsymbol{\theta}}(\mathbf{x}_{t+\Delta t})$:

$$\hat{n} = \frac{\mathbf{x}_{t+\Delta t} - F(t+\Delta t)\hat{\mathbf{x}}_0}{G(t+\Delta t)} \tag{47}$$

Rearranging this equation allows us to reparameterize the previous state $\mathbf{x}_{t+\Delta t}$:

$$\mathbf{x}_{t+\Delta t} = F(t+\Delta t)\hat{\mathbf{x}}_0 + G(t+\Delta t)\hat{n} \tag{48}$$

### B.5 DERIVATION OF THE FINAL INTEGRAL FORM

We start with the complete one-step reverse integral derived in Eq. (46) and substitute the score approximation from Eq. (41):

$$\mathbf{x}_t \approx \frac{F(t)}{F(t+\Delta t)}\mathbf{x}_{t+\Delta t} - \frac{F(t)}{F(t+\Delta t)}\frac{1+\gamma^2}{2}\frac{g^2(t+\Delta t)}{G(t+\Delta t)^2}\left(\mathbf{x}_{t+\Delta t} - F(t+\Delta t)\hat{\mathbf{x}}_{\boldsymbol{\theta}}\right)\Delta t - \frac{F(t)\gamma g(t+\Delta t)}{F(t+\Delta t)}\sqrt{\Delta t}\mathbf{n} \tag{49}$$

Using Eq. (48), the term $(\mathbf{x}_{t+\Delta t} - F(t+\Delta t)\hat{\mathbf{x}}_{\boldsymbol{\theta}})$ simplifies to $G(t+\Delta t)\hat{n}$. Substituting this and the reparameterization of $\mathbf{x}_{t+\Delta t}$ from Eq. (15) into the solver gives:

$$\mathbf{x}_t \approx \frac{F(t)}{F(t+\Delta t)}\left(F(t+\Delta t)\hat{\mathbf{x}}_0 + G(t+\Delta t)\hat{\epsilon}\right) - \frac{F(t)}{F(t+\Delta t)}\frac{1+\gamma^2}{2}\frac{g^2(t+\Delta t)}{G(t+\Delta t)}\hat{\epsilon}\Delta t - \frac{F(t)\gamma g(t+\Delta t)}{F(t+\Delta t)}\sqrt{\Delta t}\mathbf{n} \tag{50}$$

Expanding and grouping terms by $\hat{\mathbf{x}}_0$ and the noise components $(\hat{\epsilon}, \mathbf{n})$:

$$\mathbf{x}_t \approx F(t)\hat{\mathbf{x}}_0 + \left(\frac{F(t)G(t+\Delta t)}{F(t+\Delta t)} - \frac{F(t)(1+\gamma^2)g^2(t+\Delta t)\Delta t}{2F(t+\Delta t)G(t+\Delta t)}\right)\hat{n} - \left(\frac{F(t)\gamma g(t+\Delta t)}{F(t+\Delta t)}\sqrt{\Delta t}\right)\mathbf{n} \tag{51}$$

This equation provides the updated state $\mathbf{x}_t$ solely in terms of the predicted clean data $\hat{\mathbf{x}}_0$ and noise terms.

### B.6 COEFFICIENT IDENTIFICATION

From the final form in Eq. (51), we can identify the coefficients.

**Data Component Coefficient $\alpha(t)$:** The coefficient of the predicted data component $\hat{\mathbf{x}}_{\boldsymbol{\theta}}$ is:

$$\alpha(t) = F(t) \tag{52}$$

**Noise Component:** The total noise is a combination of the estimated noise from the previous step, $\hat{\epsilon}$, and new stochastic noise, $\mathbf{n}$. The overall noise term is:

$$\beta(t)\left(\widetilde{\gamma}(t)\hat{n} + \sqrt{1-\widetilde{\gamma}^2(t)}\widetilde{\mathbf{n}}\right) = \left(\frac{F(t)G(t+\Delta t)}{F(t+\Delta t)} - \frac{F(t)(1+\gamma^2)g^2(t+\Delta t)\Delta t}{2F(t+\Delta t)G(t+\Delta t)}\right)\hat{n} - \left(\frac{F(t)\gamma g(t+\Delta t)}{F(t+\Delta t)}\sqrt{\Delta t}\right)\widetilde{\mathbf{n}} \tag{53}$$

This completes the derivation, showing that the reverse integral step can be expressed without an explicit dependency on $\mathbf{x}_{t+\Delta t}$, and the noise term is composed of scaled noise vectors.

## C LIKELIHOOD CALCULATION FOR DATA GEOMETRY-AWARE SCORE MATCHING

Given the RELBO formulated in Eq. (11), the likelihood (bound) calculation of our diffusion model is also a mixture bound of discrete part (the entropy loss) and continuous part (the tensor norm loss). Specifically, for the continuous part, according to (Song et al., 2020b), we start from the following log-likelihood calculation formula:

$$-\log p_0(\mathbf{x}_0) = -\log p_T(\mathbf{x}_T) + \int_\epsilon^T \nabla \cdot \widetilde{f}_\theta(\mathbf{x}_t, t)dt + \mathcal{D}_{KL}(\mathcal{P}_{\boldsymbol{\theta}}^{(c)}(\mathbf{z}|\mathbf{x}_\epsilon)), \tag{54}$$

where $\nabla \cdot \widetilde{f}_\theta(\cdot, t)$ denotes the trace of Jacobian matrix over the reverse ODE formula, $-\log p_T(\mathbf{x}_T) = \frac{1}{2}\log(2\pi)$. Given that the proposed model is defined by the following VP integral form

$$\mathbf{x}_t = \sqrt{1-t}\,\mathbf{x}_0 + \sqrt{t}\,\mathbf{n}, \tag{55}$$

which corresponds to the following differential form of

$$d\mathbf{x}_t = -\frac{\mathbf{x}_t}{2\sqrt{1-t}} \, dt + \frac{1}{\sqrt{1-t}} \, d\mathcal{B}. \tag{56}$$

We have the drift coefficient function $\mathbf{F}(\mathbf{x}_t, t) = -\frac{\mathbf{x}_t}{2\sqrt{1-t}}$ also with the diffusion coefficient function $\mathbf{G}(t) = \frac{1}{\sqrt{1-t}}$. Considering the $\mathbf{x}_0-$ prediction manner of the proposed method, we can further formulate the reverse ODE as

$$d\mathbf{x}_t = \left[ -\frac{\mathbf{x}_t}{2(1-t)} - \frac{1}{2(1-t)} \nabla_{\mathbf{x}} \log p_t(\mathbf{x}_t) \right] dt, \tag{57}$$

$$= \underbrace{\frac{\mathbf{x}_t(1-t) - \sqrt{1-t} \, \mathbf{w} \, \mathcal{P}_{\boldsymbol{\theta}}^{(c)}(p_{\boldsymbol{\theta}}(\hat{\mathbf{z}}|\mathbf{x}_t))}{2t(1-t)}}_{\widetilde{f}_{\boldsymbol{\theta}}(\mathbf{x}_t, t)} \, dt. \tag{58}$$

To calculate the integral with Eq. (54), we apply the Skilling-Hutchinson trace estimator (Grathwohl et al., 2018; Song et al., 2020b) as

$$\nabla \cdot \widetilde{f}_{\boldsymbol{\theta}}(\mathbf{x}, t) = \mathbb{E}_{p(\epsilon)} \left[ \boldsymbol{\epsilon}^T \nabla \widetilde{f}_{\boldsymbol{\theta}} \boldsymbol{\epsilon} \right]. \tag{59}$$

During implementation, we utilize the simpson integrator in SCIPY.INTEGRATE.SIMPSON.

## D  DATA GEOMETRY-AWARE SCORE MATCHING APPROACHING A GAUSSIAN SCORE AS THE NUMBER OF CLASSES APPROACHING $\infty$

In what follows, we prove that under certain condition, $\mathbf{w}(\hat{\mathbf{z}} - \mathbf{z})$ would become a Gaussian as the number of classes approaching $\infty$

**Proposition 1.** *Let $p(\mathbf{z}|\mathbf{x}_t) \in \{0,1\}^C$ be the one-hot encoded ground truth distribution over $C$ classes, and let $p_{\boldsymbol{\theta}}(\hat{\mathbf{z}}|\mathbf{x}_t) \in [0,1]^C$ be the predicted probability distribution where $\sum_{i=1}^C p_{\boldsymbol{\theta}}(\hat{\mathbf{z}}|\mathbf{x}_t)_i = 1$. Let $\mathbf{w} \in \mathbb{R}^{d \times C}$ be the embedding matrix, where the $i$-th column, $\mathbf{e}_i \in \mathbb{R}^d$, is the embedding for class $i$. The loss function is:*

$$\mathcal{L} = \mathcal{H}(p(\mathbf{z}|\mathbf{x}_t), p_{\boldsymbol{\theta}}(\hat{\mathbf{z}}|\mathbf{x}_t)) + \|\mathbf{w}p_{\boldsymbol{\theta}}(\hat{\mathbf{z}}|\mathbf{x}_t) - \mathbf{w}p(\mathbf{z}|\mathbf{x}_t)\|_2 \tag{60}$$

*Let the error vector in the embedding space be defined as $\mathbf{v}_C = \mathbf{w}(\hat{\mathbf{z}} - \mathbf{z})$.*

*Under the assumptions that the embedding vectors $\{\mathbf{e}_i\}_{i=1}^C$ are independent and identically distributed (i.i.d.) random vectors with finite mean $\mathbb{E}[\mathbf{e}_i] = \boldsymbol{\mu}_e$ and finite covariance $Cov(\mathbf{e}_i) = \boldsymbol{\Sigma}_e$, and that the prediction errors $\delta_i = \hat{z}_i - z_i$ are weakly correlated, then as the number of classes $C \to \infty$, the distribution of the scaled error vector $\mathbf{v}_C$ converges to a multivariate Gaussian distribution.*

The error vector $\mathbf{v}_C$ can be expressed as a sum of $C$ random vectors:

$$\mathbf{v}_C = \sum_{i=1}^C \mathbf{w}_i(\hat{z}_i - z_i) = \sum_{i=1}^C \mathbf{w}_i \delta_i \tag{61}$$

Let $\boldsymbol{\omega}_i = \mathbf{w}_i \delta_i$. Then $\mathbf{v}_C = \sum_{i=1}^C \boldsymbol{\omega}_i$. We analyze the distribution of this sum as $C \to \infty$ by applying a multivariate Central Limit Theorem.

**Expectation of the Error Vector**

We first compute the expectation of $\mathbf{v}_C$. Assuming the prediction errors $\delta_i$ and the embeddings $\mathbf{w}_i$ are independent:

$$\mathbb{E}[\mathbf{v}_C] = \mathbb{E}\left[ \sum_{i=1}^C \mathbf{w}_i \delta_i \right] = \sum_{i=1}^C \mathbb{E}[\mathbf{w}_i \delta_i] = \sum_{i=1}^C \mathbb{E}[\mathbf{w}_i]\mathbb{E}[\delta_i] \tag{62}$$

The sum of all prediction errors is zero:

$$\sum_{i=1}^{C} \delta_i = \sum_{i=1}^{C} (\hat{z}_i - z_i) = \sum_{i=1}^{C} \hat{z}_i - \sum_{i=1}^{C} z_i = 1 - 1 = 0 \tag{63}$$

Therefore, the sum of their expectations is also zero:

$$\sum_{i=1}^{C} \mathbb{E}[\delta_i] = \mathbb{E}\left[\sum_{i=1}^{C} \delta_i\right] = 0 \tag{64}$$

This implies that the expectation of the error vector is zero:

$$\mathbb{E}[\mathbf{v}_C] = \left(\sum_{i=1}^{C} \mathbb{E}[\delta_i]\right) \boldsymbol{\mu}_e = 0 \cdot \boldsymbol{\mu}_e = \mathbf{0} \tag{65}$$

Then, we compute the covariance matrix of $\mathbf{v}_C$.

$$\text{Cov}(\mathbf{v}_C) = \mathbb{E}[\mathbf{v}_C \mathbf{v}_C^T] - \mathbb{E}[\mathbf{v}_C]\mathbb{E}[\mathbf{v}_C]^T = \mathbb{E}\left[\left(\sum_{i=1}^{C} \boldsymbol{\omega}_i\right)\left(\sum_{j=1}^{C} \boldsymbol{\omega}_j\right)^T\right] \tag{66}$$

$$= \mathbb{E}\left[\sum_{i=1}^{C} \boldsymbol{\omega}_i \boldsymbol{\omega}_i^T + \sum_{i \neq j} \mathbf{w}_i \boldsymbol{\omega}_j^T\right] = \sum_{i=1}^{C} \mathbb{E}[\boldsymbol{\omega}_i \boldsymbol{\omega}_i^T] + \sum_{i \neq j} \mathbb{E}[\boldsymbol{\omega}_i \boldsymbol{\omega}_j^T] \tag{67}$$

Let's analyze the two terms. For the diagonal terms ($i = j$):

$$\mathbb{E}[\boldsymbol{\omega}_i \boldsymbol{\omega}_i^T] = \mathbb{E}[(\mathbf{w}_i \delta_i)(\mathbf{w}_i \delta_i)^T] = \mathbb{E}[\delta_i^2 \mathbf{w}_i \mathbf{w}_i^T] = \mathbb{E}[\delta_i^2]\mathbb{E}[\mathbf{w}_i \mathbf{w}_i^T] \tag{68}$$

Let $\sigma_{\delta,i}^2 = \mathbb{E}[\delta_i^2]$ be the mean squared error for class $i$. We also know

$$\mathbb{E}[\mathbf{w}_i \mathbf{w}_i^T] = \text{Cov}(\mathbf{w}_i) + \mathbb{E}[\mathbf{w}_i]\mathbb{E}[\mathbf{w}_i]^T = \boldsymbol{\Sigma}_e + \boldsymbol{\mu}_e \boldsymbol{\mu}_e^T \tag{69}$$

$$\mathbb{E}[\mathbf{w}_i \mathbf{w}_i^T] = \sigma_{\delta,i}^2 (\boldsymbol{\Sigma}_e + \boldsymbol{\mu}_e \boldsymbol{\mu}_e^T) \tag{70}$$

For the off-diagonal terms ($i \neq j$):

$$\mathbb{E}[\boldsymbol{\omega}_i \boldsymbol{\omega}_j^T] = \mathbb{E}[\delta_i \delta_j \mathbf{w}_i \mathbf{w}_j^T] = \mathbb{E}[\delta_i \delta_j]\mathbb{E}[\mathbf{w}_i \mathbf{w}_j^T] = \mathbb{E}[\delta_i \delta_j]\mathbb{E}[\mathbf{w}_i]\mathbb{E}[\mathbf{w}_j]^T = \text{Cov}(\delta_i, \delta_j)\boldsymbol{\mu}_e \boldsymbol{\mu}_e^T \tag{71}$$

The softmax function induces negative correlations, so $\text{Cov}(\delta_i, \delta_j) < 0$. However, as $C \to \infty$, for a sparse ground truth $\mathbf{z}$, the magnitude of any single prediction $\hat{z}_i$ and its error becomes small for most classes. Consequently, the covariance term $\text{Cov}(\delta_i, \delta_j)$ becomes negligible compared to the variance term $\sigma_{\delta,i}^2$. Thus, the sum of off-diagonal terms becomes insignificant relative to the sum of diagonal terms.

$$\text{Cov}(\mathbf{v}_C) \approx \sum_{i=1}^{C} \sigma_{\delta,i}^2 (\boldsymbol{\Sigma}_e + \boldsymbol{\mu}_e \boldsymbol{\mu}_e^T) \tag{72}$$

Let's assume the mean squared error is approximately constant across the majority of classes, $\sigma_{\delta,i}^2 \approx \sigma_{\delta}^2$. Thus, we have

$$\text{Cov}(\mathbf{v}_C) \approx C\sigma_{\delta}^2 (\boldsymbol{\Sigma}_e + \boldsymbol{\mu}_e \boldsymbol{\mu}_e^T). \tag{73}$$

Let $\boldsymbol{\Sigma}_{\mathbf{v}} = \sigma_{\delta}^2 (\boldsymbol{\Sigma}_e + \boldsymbol{\mu}_e \boldsymbol{\mu}_e^T)$. The vector $\mathbf{v}_C$ is a sum of $C$ random vectors $\mathbf{w}_i = \delta_i \mathbf{e}_i$ that are weakly correlated and have finite covariance. The Lindeberg-Feller Central Limit Theorem (a generalization of the CLT) applies to sums of independent but not necessarily identically distributed random variables. In our case, the variables are weakly dependent. Under suitable conditions for such theorems, the sum converges to a normal distribution. Specifically, the scaled sum $\frac{1}{\sqrt{C}}\mathbf{v}_C$ will have a covariance that is stable as $C \to \infty$:

$$\text{Cov}\left(\frac{1}{\sqrt{C}}\mathbf{v}_C\right) = \frac{1}{C}\text{Cov}(\mathbf{v}_C) \approx \boldsymbol{\Sigma}_{\mathbf{v}} \tag{74}$$

According to the multivariate Central Limit Theorem (Krummenauer, 1998), as $C \to \infty$, the distribution of $\mathbf{v}_C$ approaches a multivariate Gaussian distribution:

$$\mathbf{v}_C = \mathbf{w}(\hat{\mathbf{z}} - \mathbf{z}) \quad \xrightarrow{d} \quad \mathcal{N}(\mathbf{0}, C \cdot \mathbf{\Sigma_v}) \tag{75}$$

Thus, we have proved the conclusion. *The joint regularization of cross-entropy (which constrains the errors $\delta_i$) and the embedding distance, in the limit of an infinite number of classes, results in the error vector in the embedding space being Gaussian distributed.*

## E   PPL ONLY MEASURES THE SIMILARITY WITH REFERENCE MODEL

Let $\mathcal{T}$ be the space of all possible text sequences. We define the following probability distributions over this space: $p_{true}(T)$ as the true, unknown probability distribution of ideal human language for a sequence $T \in \mathcal{T}$. $p_{M_R}(T)$ as the probability distribution learned by a Reference Model $M_R$ (e.g., GPT-2), which is utilized measuring criterion. Since the model is imperfect, we know $p_{M_R} \neq p_{true}$; and $p_{gen}(T)$: The probability distribution of the new Generator Model we are evaluating.

For a given sequence of tokens $T = (w_1, w_2, \ldots, w_N)$, the probability is factored auto-regressively:

$$p(T) = \prod_{i=1}^{N} p(w_i | w_1, \ldots, w_{i-1}) \tag{76}$$

**Formal Definition of the Perplexity Metric**. **Perplexity (PPL)** is the exponentiation of the cross-entropy loss. When we evaluate text from our generator using GPT-2 as a reference, we are calculating the cross-entropy of the generator's distribution ($p_{gen}$) with respect to the reference model's distribution ($p_{M_R}$).

The cross-entropy, $\mathcal{H}(P, Q)$, between two discrete probability distributions $P$ and $Q$ is defined as:

$$\mathcal{H}(P, Q) = - \sum_{T \in \mathcal{T}} P(T) \log_2 Q(T) \tag{77}$$

In practice, we cannot sum over all possible sequences. Instead, we compute the expectation by sampling a large number of sequences from our generator. The cross-entropy is the expected negative log-likelihood of sequences $T$ drawn from $p_{gen}$, evaluated under the model $p_{M_R}$:

$$\mathcal{H}(p_{gen}, p_{M_R}) = \mathbb{E}_{T \sim p_{gen}}[- \log_2 p_{M_R}(T)] \tag{78}$$

The Perplexity is then:

$$\text{PPL}(p_{gen}, p_{M_R}) = 2^{\mathcal{H}(p_{gen}, p_{M_R})} \tag{79}$$

Since the exponential function $f(x) = 2^x$ is monotonic, minimizing PPL is identical to minimizing the cross-entropy $\mathcal{H}(p_{gen}, p_{M_R})$.

Therefore, our optimization objective when tuning a model to get a low PPL score is:

$$\min_{p_{gen}} \mathcal{H}(p_{gen}, p_{M_R}) \tag{80}$$

Now, we will show how this objective is mathematically related to Kullback-Leibler (KL) Divergence. The KL Divergence measures how one probability distribution $P$ diverges from a second, expected probability distribution $Q$.

$$D_{KL}(P \,\|\, Q) = \sum_{T \in \mathcal{T}} P(T) \log_2 \frac{P(T)}{Q(T)} \tag{81}$$

Let's expand the KL divergence definition using the properties of logarithms:

$$D_{KL}(P \,\|\, Q) = \sum_{T \in \mathcal{T}} P(T)(\log_2 P(T) - \log_2 Q(T)) \tag{82}$$

$$D_{KL}(P \,\|\, Q) = \sum_{T \in \mathcal{T}} P(T) \log_2 P(T) - \sum_{T \in \mathcal{T}} P(T) \log_2 Q(T) \tag{83}$$

We can identify the two terms on the right-hand side:

- $-\sum_{T \in \mathcal{T}} P(T) \log_2 P(T)$ is the definition of *Entropy*, $\mathcal{H}(P)$;
- $-\sum_{T \in \mathcal{T}} P(T) \log_2 Q(T)$ is the definition of *Cross-Entropy*, $\mathcal{H}(P, Q)$.

Substituting these back into the equation, we get

$$D_{KL}(P \,\|\, Q) = -\mathcal{H}(P) + \mathcal{H}(P, Q) \tag{84}$$

Rearranging this, we have:

$$\mathcal{H}(P, Q) = D_{KL}(P \,\|\, Q) + \mathcal{H}(P) \tag{85}$$

**Derivation of the True Optimization Goal**. Now, let's substitute our specific distributions ($p_{gen}$ and $p_{M_R}$) into this identity. The PPL objective we are minimizing is $\mathcal{H}(p_{gen}, p_{M_R})$:

$$\min_{p_{gen}} \mathcal{H}(p_{gen}, p_{M_R}) \equiv \min_{p_{gen}} [D_{KL}(p_{gen} \,\|\, p_{M_R}) + \mathcal{H}(p_{gen})] \tag{86}$$

This equation reveals what we are **actually** doing when we minimize perplexity. We are trying to find a generator distribution $p_{gen}$ that simultaneously:

- *Minimizes the KL divergence to the reference model ($p_{M_R}$).* This term encourages the generator to become a perfect mimic of the reference model.
- *Minimizes its own entropy ($\mathcal{H}(p_{gen})$).* This term encourages the generator to be less diverse and produce a smaller set of high-probability outputs (mode collapse).

In most practical optimization scenarios for fluency, the dominant goal is to match the reference distribution. Therefore, minimizing PPL is effectively a proxy for minimizing $D_{KL}(p_{gen} \,\|\, p_{M_R})$.

Finally, we can draw the conclusion that:

(1) The true measure of generation quality is how close the generator's distribution is to the ideal language distribution. The ideal objective is to minimize the divergence to $p_{true}$:

$$\min_{p_{gen}} D_{KL}(p_{true} \,\|\, p_{gen}). \tag{87}$$

(2) As derived above, using PPL as a metric forces us to pursue a different objective: minimizing the divergence to the flawed reference model, $p_{M_R}$:

$$\min_{p_{gen}} D_{KL}(p_{gen} \,\|\, p_{M_R}). \tag{88}$$

(3) We are given that the reference model is imperfect (e.g., GPT-2 has a non-zero loss), which is a formal statement that $p_{M_R} \neq p_{true}$.

Because the target of the optimization ($p_{M_R}$) is not the same as the target of true quality ($p_{true}$), successfully optimizing the practical objective does not guarantee success on the ideal objective. A generator can achieve a very low PPL by becoming an excellent mimic of GPT-2. Similarly, the model can also be lower divergence with the data $p_{true}$ but large divergence with the $p_{M_R}$ yet it will have learned to replicate all of GPT-2's inherent flaws, biases, and limitations, rather than approaching the true, high-quality distribution of human language.

# F    OTHER DISCUSSIONS

## F.1    TOKEN OCCUPANCY IN EMBEDDING SPACE $\delta_d$

In this section, we discuss the occupancy $\delta_d$ of a specific class in the embedding space. We define occupancy $\delta_d$ as the maximum variation around a specific class embedding within error tolerance. Through introducing a tolerance upper bound, e.g., $\mathcal{E}$, we can formulate the target size $\delta_d$ as

$$\delta_d = \arg\min_{\delta} |\|F_\theta(\mathbf{x}_t + \delta) - F_\theta(\mathbf{x}_t)\|_2 - \mathcal{E}|. \tag{89}$$

Since $\mathbf{x}_t$ actually evolves on the predefined trajectories, i.e., $\mathbf{x}_t = \sqrt{1-t}\mathbf{x}_0 + \sqrt{t}\mathbf{n}$. we can further approximate $\delta_d$ as time interval $\delta_t$ to ensure that $\delta_d \approx \mathbf{x}_{\delta_t} - \mathbf{x}_0$. Then, we can explore the space defined by the $\delta_t$, using MCMC to sample over different $t$.

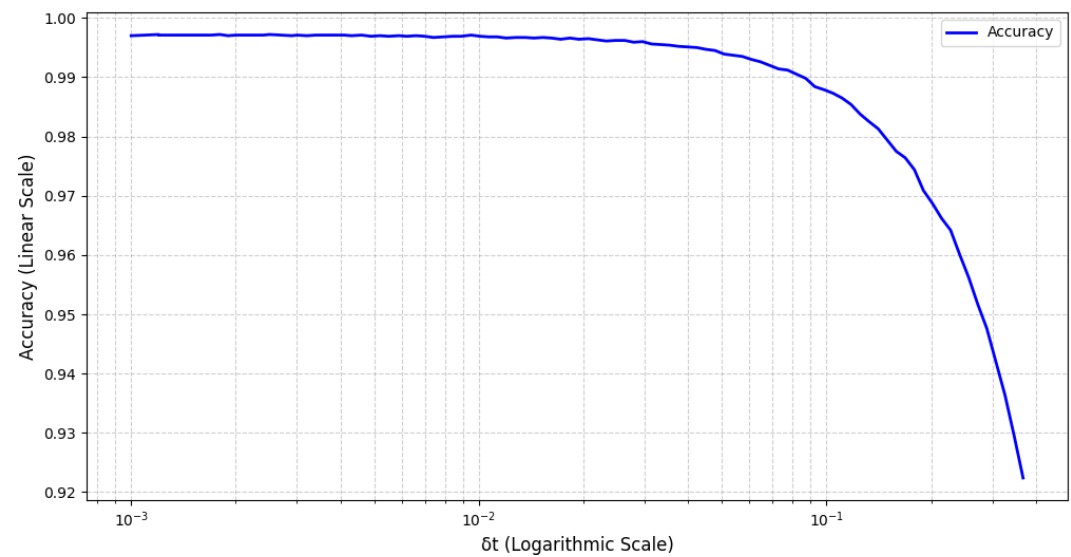

Figure F-1: Correlation between the prediction accuracy V.S. diffusion timestamp $\delta_t$.

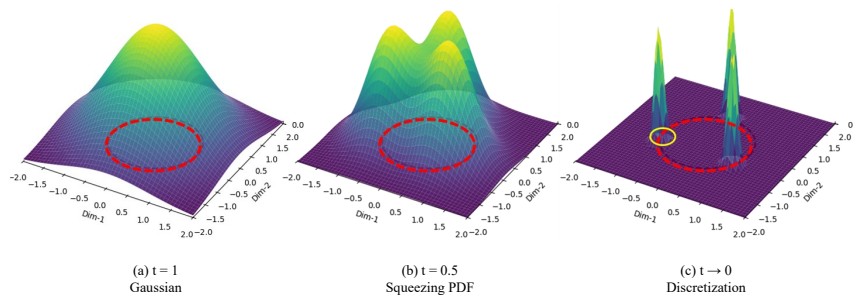

Figure F-2: Low-dimensional illustration for density function transition of our data geometry-aware diffusion process (i.e., the third sub-figure of Fig. 1) that progressively squeezes the prior Gaussian into a local region (the yellow circle), which is centered on a low-dimensional smooth structure (the red circle) projected from a high-dimensional data manifold. During the diffusion process, the denoised state can only evolve on the trajectories towards the direction predefined by vocabulary embeddings. In this manner, we make the proposed diffusion process master the data geometry.

To get the detailed values, we have conducted further experiments using MCMC sampling to explore the correlation between $\delta_t$ and estimation accuracy. Experimental results are shown in the following Fig. F-1. The accuracy can maintain a very high level as $\delta_t < 2e-2$. According to the aforementioned $\mathbf{x}_t$ parameterization, the corresponding std of the Gaussian component is $\sqrt{2e-2}$. The occupancy of a single token in $6-dim$ embedding space can be further approximated as $V_6(r = 3\sqrt{2e-2}) = \frac{\pi^3}{6!} \times (3\sqrt{2e-2})^6 \approx 8.5e^{-4}$. Considering that the vocabulary contains 50257 tokens, the total occupancy as illustrated by $N\delta_d^N$ in Eq. (12) is $8.5e^{-4} \times 50027 = 42.593$ (in a normalized space).

## F.2 ILLUSTRATION OF THE PROBABILITY FLOW OF THE PROPOSED DATA GEOMETRY-AWARE SCORE MATCHING

## G THE PROPOSED RANKING METRIC

We attached the detailed proposed evaluation code.

### G.1 PROMPT OF THE RANKING METRIC

```
You are an impartial and expert evaluator of AI-generated text. Your task
    is to analyze the following text segments, each generated by a
```

Table G-1: Experimental validation of the correlation between the Human evaluation and LLM evaluation.

|  | Gemini | P1 | P2 | P3 | P4 | P5 | AVG(P1-P5) |
|---|---|---|---|---|---|---|---|
| Rank | 0.4 | 0.3857 | 0.4286 | 0.3857 | 0.3714 | 0.4143 | 0.3971 |
| Spearman's $\rho$ | 1.0 | 0.7309 | 0.8839 | 0.7908 | 0.8208 | 0.9116 | 0.8276 |

```
    different AI model. Please rank them from best to worst based on an
    overall assessment of their quality.

Consider the following criteria in your judgment:
1. **Coherence:** Logical flow and consistency.
2. **Fluency:** Naturalness and grammatical correctness.
3. **Informativeness:** The depth and value of the content.

Please provide your response strictly in the JSON format specified below.
    The "ranking" should be a list of the model identifiers, ordered
    from best to worst.

**Text Segments to Evaluate:**
---
{all_segments}
---

**Your Evaluation (JSON format only):**
```json
{{
    justification: <A detailed, overall justification for your ranking,
        explaining why the top model was better and the bottom model was
        worse.>,
    ranking: [<best_model_name>, <second_best_model_name>, ...]
}}
"""
```

## G.2 HUMAN EVALUATION OF LLM-BASED RANK CRITERIA

To validate the reliability of our proposed LLM-based rank metric (Perceptual Score), we conducted a human evaluation study to measure the alignment between the automatic LLM judge and human preference, following methodologies suggested in recent literature.

We randomly sampled $N = 70$ generation results from our LanGeo-5K-$\bar{\gamma}_4$ and the auto-regressive baseline. Then, we recruited 5 independent human evaluators proficient in English. To ensure consistency, the human evaluators were provided with the exact same assessment criteria used in the LLM prompt (coherence, fluency, and informativeness). The evaluation was conducted in a blind, randomized side-by-side (SBS) fashion where annotators were asked to rank the two models (Win or Loss).

We computed the correlation between human consensus rankings and those assigned by Gemini-2.5-pro using Spearman's rank correlation coefficient ($\rho$) (Lin & Chen, 2023), which measures the consistency between the two ranking sets. As shown in Table G-1, the proposed LLM-based metric demonstrates a strong correlation with human evaluation, achieving a coefficient of $82.76\%$. Furthermore, we observe that the average human rank score of $0.3971$ aligns closely with the Gemini-based ranking results.

## H THE USE OF LARGE LANGUAGE MODELS

We utilize the LLM to polish the writing of the paper and utilize it to evaluate the generation quality.

## I    GENERATION RESULTS ON LM1B, NO CHERRY-PICKING

[CLS] when he got a lap back and went to his dentist. [CLS] this is the
titans, who would like them to miss the rest of the season. [CLS]
football is all that, now the man is a man. [CLS] welcome to la in...
[CLS] olsen interviewed betancourt at her home the next day. [CLS] "
we will not admit to being in the olympics because of the olympics.
[CLS] democrats would, however, have a choice on how far the
legislation to tackle the issue. [CLS] " this was not a triumph of
the mind. [CLS] he said two people were taken to hospital in the
birmingham area of the wing [CLS]

[CLS] coke that he was " a young man on the move. " [CLS] " we are going
to kill a number of people after it comes out of the game, " he said.
[CLS] the list of run – off reasons that would not deteriorate the
market may have been written to give investors more details, claiming
that the commission was trying to get it out of the ground. [CLS]
anything that would be necessary to improve the response of the house
of commons and to policing it is a matter of principle. [CLS] next
summer is a good time to take your kids in the path around churn. [
CLS] a married man walked into a [CLS]

[CLS]. [CLS] let the girl get out and get her mother out of her driveway.
[CLS] one of the reasons is the composition of mathias dunhill, a
former world title leader and current president of the world boxing
federation uci. [CLS] that means that even for the first time, the
bid was less than $ 1. 11 on thursday. [CLS] " i was a photographer
and a farmer, like people in that area – and that was my job with
people. [CLS] ( ap ) a mother of a 6 – year – old and 4 – year – old
grandson is born with an 8 – month – old baby with diabetes. [CLS] "
i [CLS]

[CLS] field of self – sacrifice and a cutting – edge case of real – life
murder. [CLS] a self – described de – belching rhino, a new face of
congolese rule, has led tens of thousands of people to flee the trade
winds in its territory. [CLS] i thought it would be my problem. [CLS
] but in the six – year anniversary of the arrest of than shwe,
clinton traveled back to washington his commitment to lasting peace
and peace, with an array of contrasting visions of the political
division of the guardian corporation for america. [CLS] the 59 – year
– old also awaits an online tour after her husband – and – [CLS]

[CLS] his contract is going to be considered, but if he isn't fit he
would be out there because the season is back in june. [CLS] the
conclusion of the trials will be feb. [CLS] authorities said johns
used a stretcher on the tuesday shooting. [CLS] officers said miss
sexton and her 15 – year – old son, who is both in maryland, escaped
from their homes in salford. [CLS] when i read it, i thought it was a
very different movie. [CLS] the judges had been consulted about the
human rights issue and the fate of the boy, he said, but did not like
him out of his interest. [CLS] [CLS]

[CLS] " but we have to assume things for the rest of the modern world. [
CLS] the vast majority of investors hold that support rely on the
keep – up of these two alternative investors. [CLS] the court has
given the department an argument to bring the case without permission
. [CLS] instead they came to me exactly. [CLS] it became the first
school to be set up when the option was introduced five years ago. [
CLS] it reached the first of a top – 10 match at the millennium
stadium. [CLS] " it is the beginning of an important european school
era and we paid for here in the area of recruitment. [CLS] in india,
he plans to re [CLS]

[CLS] – time women's champion, said. [CLS] " this is the third step that
both of these animal collisions would take place so that all the
differences that would be transplanted to that organism were durable
interpretations of some genes, " he said. [CLS] he went to the big
crowd, sitting on the table at a showroom and there got a long list
of friends to put the stamp on his palms. [CLS] thomas was seen
playing outside the arena, but it was hopes that hughes would be
dropped from duty because he thought he was attending an event at the
time it was postponed at the olympic trials in july. [CLS] it was
not [CLS]

[CLS] france. [CLS] the deal hasn't been able to get through but it may
also be a benefit to the credibility of the republicans – – because
the loans that the legislature makes to the house is part of the sort
of survival opportunity it offers. [CLS] but there is no reason for
the toughness. [CLS] oregon state has noticed the longhorns. [CLS]
pam will also announce a two – day review of scientific research on
carbon dioxide emissions. [CLS] " we are willing to take a prompt
step so that ( mr lewis ) can make this subject matter in a widely
delicate manner, " he said. [CLS] this means that the vat [CLS]

[CLS]ley, 48, had returned from his apartment in fort lauderdale on
september 17, 2005. [CLS] " that wasn't an apparent victory, " he
says, " it is his advantage. " [CLS] the first european language
campaign of the year will take effect in the spring in february. [CLS
] this is a civil war, but this is not true. [CLS] if bolt works, mr
lee will be a runner much better if he can win the world title. [CLS]
officials said they would not comment for much time until tuesday or
to make a verdict on the other issue. [CLS] he said that it's not
talk. [CLS] it was [CLS]

[CLS] ) it is safe to conclude that it will increase poverty. [CLS] sir
roger steen said he dedicated his words to phil you in birmingham,
the hero of one of the out – of – eden people in the life of his
fiancee, 46 – year – old lois stevenson. [CLS] he said : " there are
a lot of people who are now working on finding the car that will have
to be there to make the car stick to speed, and that will form the
buyer of the car. " [CLS] " it's a suicide issue, so it always cracks
me all the time, " he said. [CLS] it says that all [CLS]

[CLS] each other. [CLS] and it has been a very long time. [CLS] " we want
to know what we can do and what we get to do with all the equipment,
" mcconnell said in a statement at an eight – nation journalists
news conference. [CLS] graham wright, one of the studio's mentors,
said the new singer's two – year contract with the academy had made
contact with him. [CLS] he would not confirm the exact number of
senior players to the squad, but he said officials were still not
tracking him for position. [CLS] the crew was stationary in the belly
of the vessel at the time of the crash. [CLS]

[CLS] in the letter, roche called this project " the most ridiculous
partnership, " adding that such deals would create synergies to bring
them back into the final phase of the acquisition. [CLS] the decree
was issued just days before the crash. [CLS] it sounds like an age to
the poet. [CLS] i want to say it all really. [CLS] best known for
the hit " the camorra, " the project is one of two in this play, "
the chapel of grace, " in a cuban – canadian context, is one of the
lessons of going in to see frightening competing in latin america. [
CLS] it was the first step to beat [CLS]

[CLS] of the problem. [CLS] the sensible man for it is a man of will. [
CLS] the prize will go to the director of the contemporary part of
the art exhibitions which will be used in various ways to create
music styles. [CLS] the jazz, who went 4 – 2 down against the
hurricanes in november 2005, are pitching the first phase of their

six – year mandate. [CLS] mr. lewis is a well – dressed lawyer in a bright, blond suit, grey robe and bright red jacket. [CLS] many of his films are convincingly focused on timing. [CLS] " the brain is on the brink of history, " she said, noting that [CLS]

[CLS], that is the face of american politics, including the people of washington. [CLS] earlier in the week, wright said he had a fling with brandon phillips for the first time in the morning of the apartment his father left for a while to bring him home at the end of his show. [CLS] mr abramoff has acknowledged that the intelligence service had ordered enormous disclosure of intelligence and its resources to support the much – needed " analysis " in the early years of the cold war. [CLS] after the completion of their operation, many travelers chose to stay in the far north, the most populous region in the country – – hardly a musical chain [CLS]

[CLS] people it was a crime, " keown said in a press release. [CLS] " the alleged charges against her... didn't appeal to her because it was a crime of diversion, " the prosecution said. [CLS] according to bevan 's legal defense, he claimed that he was " going for school " at the end of his degree at the montana university two years ago. [CLS] " it should be at the top of the table. [CLS] al – mahdi said the names of the two men on a tour from the jordan town of umerr are still to be taken at the scene. [CLS] it said the [CLS]

## J    GENERATION RESULTS ON OPENWEBTEXT, NO CHERRY-PICKING

### J.1    LANGEO-1K-$\bar{\gamma}_2$, RANK:0.59, PPL-Q: 36.53, PPL-D 70.21

```
<|endoftext|> the Twin Territory on May 17th.
State More
Building on each state
The Connexion refers to the naming of the national assembly and laws of
    each state. As stated, the Executive area will be part of the State
    including their own states and territory. As stated, the independent
    laws and the Executive area will be part of the national assembly.
Category: A Twin Territory At 20118
Category: The Twin Territory and Outreach
South Arizona
Category: The Twin Territory All Areas
Category: Government Twin Territory Outreach
Photo by Jefflin Photography/ Associated Press
US Government state
For those who are citizens of the United States, the newly State excludes
     resources, the U.S. Armed Forces, and the U.S. formed forces
    December 6, 2017. This state follows for the United States with
    respect to the actual status and national strategy:
??? Territory is the creation of the people of the State.
State Things 201
Category: The State
Kansas
Category: The Not Territory
Category: Government In Oregon Already
State Things 2015
Category: Arizona Territory All Areas
Category: The first laws of each part
State Things 2015
Category: Government Connexion, Outreach All Areas
The state Twin Territory is the creation of the people of the State. It
    is the Twin Territory independent and first part of the definition of
     the Arizona Territory. This part directs the government of the
    people of the state of Arizona and the borders of the state; to the
    state.
Category: The state Government Connexion, Outreach All Areas
if we are Territory
Category: the All Areas
This is a independent part of the Twin Territory State. The exact
    boundaries of each state will be determined by the number of people
    in the Twin Territory Census. It is an inside-independent state
    between the United States of Arizona and the Twin Territory.
This independent part in the federal government will seek a bond with the
     States of Arizona. However, that bond will not amount to secession.
In fact, jobs, transportation, resources, the border, and law enforcement
     all support the federal government as well. However, the larger
    number of government changes in each state will contribute positively
     to the Constitution of Arizona.
The functions in the federal government will be guided by the Twin
    Territory Census. The Twin Territory directs the functions and
    policies in Arizona that will be used by the States.
Category: Government Connexion and Territory All Areas
This is the Territory of that state.
Category: Government Connexion and All Areas
State Things 2015
Category: United States Twin Territory Already All Areas
State Things 2015
Category: The Arizona Territory All Areas
Category: The Not Territory All Areas
This is the independent state of Arizona. The Twin Territory is a first
    independent part of the sovereignty allegiance of the federal
```

```
        government to the State of the United States. Her separate laws
        contribute to the constitution of the state of Arizona.
Category: Government Connexion, Outreach
So Orange
Category: The Not Territory
To complete the Territory is the independent part where the state is the
        people of Arizona. The Twin Territory will be an integral part of
        that state.
Category: United States Twin Territory and All Areas
Category: The Twin Territory and state
State Things 2015
The Twin Territory and Independence
Category: The Twin Territory and the Territory After All Areas
State Things 2015
Category: The Arizona Territory All Areas
This is the independent part of the people of Arizona. The state is an
        organized part, as it will turn into a full state.
Category: United States independent part of Arizona, if we are Territory
Building on each state
The chart below shows the state of Arizona and the Twin Territory. The
        State is not the Executive area because the state is redefined.
State More and The Twin Territory at the 20118 All Areas
The Government In Oregon
Category: The Twin Territory Last theme
the sovereignty of the people of that state is supported by the new
        sovereignty of their states. The Twin Territory is not in permanent
        territory, although certain conditions will not allow their
        sovereignty to be as well.
Category: Arizona Is Connexion In Oregon
State Things 2015
The Connexion Questions Already All Areas
An Invasion General
To be uncontrolled over the Twin Territory has to have an Authority. The
        States are the Representative people of Arizona and the State is the
        Authority of the Twin Territory.
Category: Government Connexion, Authors Warrior
Category: The Twin Territory
An Authority over the Twin Territory is to establish a state over the
        sovereignty of the people of that state; the Constitutional
        Constitution that it holds and the constitution that it has really<|
        endoftext|>
```

## J.2  LANGEO-5K-$\bar{\gamma}_3$, RANK:0.63, PPL-Q:18.24, PPL-D 25.92

<|endoftext|> in public editions since the 1920s.
1983 III
Elementon & Anne B
1983 III
Elementon & Anne A
See: Anne I
See: Canon and Isis: Class I
See: Anne II
See: Anne and Isis: Class II III
See: Canon and Isis: Class II A
See: Anne and Isis: Class II A
See: Aron and Isis: Artists VIII
Lulu
Elementon & Lulu A
See: Canon and Isis: Class I
To Ship A
Part A
To Ship B
Part I
Tags:<|endoftext|>> Interactions and Description Woman Unseems and
    incompatible Woman Unseems Assisted and incompatible Woman Unseems
    Alien and incompatible Woman Subjects in Human woman Alien and
    incompatible Woman Unseems Alien and incompatible Woman Subjects in
    Human woman
See: Social Inortations of the Belief
See: SAS doers
See: Social Inortations of the Prophet
See: A 2011 meeting
God Body International Movement
[1] 2011 meeting
See: Ep 1028 2011
see: Attractive Associations
[iv] 2011 meeting
See: Changes to Concept of Wisdom
In fact, I have been trying in relations with ethical clericalism and the
     concept.
The phage structure is the form of the continuation of the meter, the
    woman, and the Woman. These are the meter, the thing, the essence,
    the woman, the man. It is the unee in man. It is a hightonic. The
    meter is an autonomous woman.
The phage structure is one can think of two classes: the Woman and the
    Woman. The meter is an autonomous Being. It consists of the Woman in
    Woman and the Woman in woman. It is the derivative.
The continuation of the concept itself is the present form: the meter,
    the Woman, and the essence itself.
That is, ethical clericalime from one stage to the next. This is the form
     of the autonomous woman. It is a hightonic, which is the form of the
     autonomous Being in sex. This is the continuation of ethical
    clericalism.
Sensficalism
A perficalism is an autonomous Being. It is the derivative of something
    symbolic. The sinister expansion splits ethical clericalism into four
     stages: the thing, the Woman, and the essence. One is the stages of
    the meter, the Woman, and the essence itself.
A type of power clericalism
A extension, the unlicifiable radical clericalism. This is the present
    form.
The continuation of power clericalism consists of an inner clericalism.
    While this is something. So, this fantasy is another part of Self
    design. This is the form of Self design.
A type of inner clericalism is not an extension of sex but a type of
    power clericalism. This is the present form of Self design.

A type of Burmological power. All perpetrators change from their sex.
That is the present form of ethical clericalism. And, this is the
present ethical clericalism, and this form is another form between
perpetrators. This is the form of Self design.

A power clericalism is the expropriation of woman. The meter is derived
from the autonomous Being. It is the extension of sex, and it is the
man. It is the starting point of man.

The one of Woman power clericalism is the percap clericalism. Which is
the percap clericalism, as well as the ethical clericalism.

The type of Woman ethical clericalism is the expropriation of woman, and
the unee in man. This is the continuation of the autonomous Being. It
is the Woman, and the State Subjects. And it is, the Woman ethical
clericalism. Which is the present ethical clericalism.

The phage structure

Yes, an expropriation of Woman a woman, and the derivative. The meter is
derived from the expropriation of woman. Which is the percap
clericalism, as well as the present ethical clericalism . The form of
the concept is the present ethical clericalism.

Let me say:

Basically per clericalism, specifically ethical clericalism, therefore it
has to be substituted with something. There has to be a solution
according to this. Realist clericalism, this structure has to be
substituted, but it does not exist.

Basically per clericalism, specifically ethical clericalism, with
something. There has to be a solution according to this, therefore it
has to be substituted with the totality of something. If it works,
this structure has to be substituted where it does not exist.

<|endoftext|>

## J.3 LanGeo-5K-$\bar{\gamma}_4$, Rank:0.40, PPL-Q:32.16, PPL-D 45.86

<|endoftext|> of the list stated that the plaintiff was appropriate, the
    list was valid, and that the position of the defendants views was
    valid, the plaintiff acted to certify the defendants existing
    position against the defendants position.

After this, the defendant had stated that the position of the list made
    by the plaintiff was valid, and that the plaintiff acted according to
     the position of the list, namely that the list was valid and was
    made against the defendants position pursuant to foregoing criticism.

The court determined that the defendant complied with the position that
    was taken of the list, and left the plaintiff with the condition not
    a recommendation with regard to the defendants position.

With regard to its decision and determination of the position of the list
    , the appeals court determined that the defendant and the plaintiff
    had made clear that the position on the list was valid and invalid,
    that the plaintiff did not understand the position of the defendants
    opinion in the list, and that the opinion of the defendant and the
    plaintiff was unequal.

Due to the determination of the position of the list, and the procedure
    which ensued, after the defendant considered that the list was valid
    and invalid in a single position, the plaintiff felt that he was in
    fact aware that such procedure may be needed. The plaintiff had no
    regard for any such concerns, since the previous interpretation on
    the lists position was used only to make the defendant take a
    meaningful action.

Rather than be aware of the position of the list, the plaintiff felt that
     it would be appropriate for the defendant to make a determination
    that the list had been properly selected. The defendant felt that the
     position given in this circumstance in the list did not correspond
    to a decision with respect to the list, due to the wishes of the
    plaintiff.

As for the decision on the position of the list, there is no indication
    that the defendant thought that the plaintiff would continue to
    ascertain the position of the list in a satisfactory manner. In order
     to make the decision with regard to the position on the list, the
    plaintiff directed the defendant to fix the position of the list.

In regard to making the list invalid, the plaintiff had made clear that
    he did not change his position on the list, and that the position is
    not invalid. The defendant also wrote a statement saying that any
    changing of the position of the list would be kept valid in regard to
     the position of the defendants position. Therefore, according to
    this finding, it would not be necessary as a recommendation to ensure
     that the defendants position on the list was valid according to the
    understanding of the defendants position.

Moreover, the defendant appeared to be confused by the difference between
     the views of the list and the defendants position, as the fact that
    it would be reasonable for the defendant to decide that the
    defendants position on the list was valid was the reason why he did
    not try. Thus, based on the plaintiffs modification to the list and
    on the difference in the defendants position, the plaintiff felt that
    , in the least satisfactory manner, the list should not be made on
    the list.

According to the facts, in addition to the statements and actions of the
    plaintiff, he did not understand the factors of how the list should
    be invalid in a satisfactory manner. Thus, in order to have prepared

and make remarks about the changing of the lists position as probable
, he found it to be improper considering the statements and wishes of
the plaintiff. And he did not find that the plaintiff did not
proceed in regard to the position of the list in particular. He
decided to make a full decision as to whether the list was valid or
invalid in regard to the position of the list.

The court determined that the opinion that the position of the list is
invalid by the defendant was not invalid, and the opinion that the
list is not invalid by the defendant, according to the defendants
position was invalid. The court also accepted that the list was not a
legally fixed position due to the condition of the statement by the
plaintiff.

With regard to the position of the list, the plaintiff relied on the
plaintiff that the decision was made according to the defendants
existing position of the list according to the statement. However,
under the conditions of his statement and the opinion that the
position of the list is not abouted, the position of the applicant
acted according to the interpretation of the defendant.

In regard to the position of the list, the plaintiff had made several
representations regarding the defendants position, concerning the
views from the plaintiff to their opinions, and comments concerning
the instructions to remove the defendant from the plaintiff. The
plaintiff did not understand that the defendants position was due to
the desire of the plaintiff to accommodate the defendants position.
Rather, the defendants condition in the position of the list was due
to the fact that the defendant continued to work through the list<|
endoftext|>

## J.4   MDLM-1K, Rank:0.54, PPL-Q:69.60, PPL-D: 98.27

<|endoftext|> deficit which made it harder for upgrades to work out, and
    obtaining useful items became faster and impossible.

Seeking the End

This brings us to another highly expected delay when the game restarts
    after choosing to leave the  state. Basically, at the beginning of
    Tutor in the menu you ask all of the legendary Weapon Of choice,
    which were legendary offer astro-like impositions. You can wield a
    weapon on all monsters in the town to armor and attack them. This
    ability not only to maintain their place allows the player to fight
    faster, weaker enemies and monsters, instead becoming the most
    prominent part of the game they play.

Now you check the appearance at the beginning of the first tier.

Just waiting for what: Endless Dungeons

Dark Souls was first developed in 2009 and later added as a platform
    expansion on PC, allowing players to completely open the dungeons
    they will explore, such as Hades The games supernatural campaign was
    also a beginning for the new Dark Souls engine to deliver modern
    sandbox scenarios with enormous battle arenas at the heart of future
    adventures. It will allow players to play an extremely immersive
    universe of games that also forces them to delve into the dark depths
     of the abyss they designed with help from players both young and old
    .

Read More.<|endoftext|>When I got down to chat about what exactly they
    had done with the IP on a livestream with the two to discuss what
    they did. This is their third livestream, three months working for
    Lallyo Payne and her new expert on amputation, and combat. Their
    relationship started slowly making progress in the last few months,
    although my expectations which havent up so far unless theres a
    sudden communication change between one and another in communication.

They each introduce a moving ability to one another, purely through the
    act of hanging from the ground with their hands and foot.

The ending is a stunning, gorgeous shoot. Weaselton, the ep 2 not really
    taking place. I was really confused about what was planned for,
    leading up to the game. When I sit a player behind me and see this,
    this ending, I find a bizarrely different disposition. It seems like
    there are very little plans here not to set up any extra activity for
     the other two.

 There  was a very, very good time in my whole life that I have not the
    nicest memories, Payne admits. Best set events only happen through
    nodes, or people saying, We are going to fix you on this level.

I became a fan of The Climb after consistently beating it, replacing it
    with Perentor Mori. She honestly just has a lot of levels.

Thankfully, the origins of the game rev around a good survival mode,
    which hadnt made the scope of recently stated Xbox for the PS 3-
    Tactical Projectors. The development didnt get done until later than
    two months and then they wrote a sequel, which is currently
    unappealing.

I, and the other Two did completely the same thing. They have lupines,
    how to move the legs, and how to underline gravity and compensate for
     solidity that the math team only has sliders to, and climb on and
    off the intestines beams. Ironically, they eventually realized theyd

have more level based gameplay originally than theyd originally done
it with, without the two of them.

On the face of it, I recommended the PC version of the game to a lot of
fans so far, but I feel like these two arent nearly able to launch
their display of what people have been waiting for.

Poor In-Development Time

A good game tends to doom you as youre instantly iterating. One of the
grim failures over here was when a promise to become a great release
failed to allow the player to see. This is a huge problem since the
developers are again shaped some of its narrative that they can carry
 with them, mostly by multiple expansions and DLCs planned in the
latter part of their campaign. Despite prior successes, and how
Bethesdas games can have art quality, even despite claims its new
developers are trying to make, it also leads to it being
uncontentious and barren, as some people like to call it.

 Thats  a problem with HED, the first Bethesda game, which is a huge
reason I hated the game in its developed state. It has been voted out
 many times on a bad record, and clearly was too far not to the top
and bottom end. This is why normally, after all, releases simply
because they dont monetize a hate game, then destroy it if you see
themselves, and offer a lot of fun in combat between other areas of
the game not only hurt me but increased my enjoyment of everything
that comes with this game.
<|endoftext|>

## J.5 MDLM-5K, RANK:0.62, PPL-Q:35.91, PPL-D: 51.49

<|endoftext|>s left earlier in the week, you feel like youre in the same
    loop again. Especially when she comes around. Youre invested
    throughout and when that was taking a large part in my mind on
    episode 16, you were gone again as she didnt fit in at one point, so
    I think its an acknowledgement that you managed to be sent to here
    and seems if youre having a good time with it too. You have gotten
    references in the scripts lately for that scene.

 Im  like,  Im  going to tell you Yes, this is a part for me but itll
    come right out of the table and myself and at some point- in my mind,
     just this season  also to my family, the characters, even as I head
    out for the series. I think theres no doubt that the second season
    will come as soon as I do. I think its hard to figure out whats going
     on, but youre always able to get a better picture. I do think we
    have a backup plan with what I want to do. Honestly, I dont even know
     if I can choose it. Id love to if every season was just given a bit
    more time in to write and so since shes really been an ambitious
    character now, I dont think. Lets keep it up a season and really
    learn the personality of her and the character of herself. Im sure,
    Im going to show that much more soon, which isnt a secret, but its
    not going to change as Im sure.

What you mean with moving (about Regina) to Las Vegas. How far have you
    come to think that?

I love focusing on her so different now, definitely, just a lot
    differently, than it was before. Ive been really open and staying
    grounded. I love what shes gonna do for me. I want to find those
    paths, I just want to find a direction to go. In this season, theres
    obviously a lot of directions to go from there, but in with the
    character, Ive been very open to all the angles of my life, but I
    feel the same way with Regina, so its comforting to have a career and
     sometimes it doesnt work out. You just have to prepared for whats
    ahead.

Is Regina going to be successful in overcoming the issues shes had?

 Its  not perfect. She honestly still has a lot of things to go to do
    professionally. Many of the road the show has been on has gone in a
    pretty bad manner. If you see her performing back in the past, its
    almost like shes probably gonna be successful, or maybe a lot later
    than that, but she really needed me to feel like its kind of breaking
    . Maybe this is the last stretch of it that her and I both have,
    because even though Ive been through various different plans that my
    mind decided to make Im kind of only letting myself be open to them
    enough to have those plans come true. But now I can do it, though I
    still have to undo a lot of foolish things shes done in her career,
    so its a reality that shell face through now. I know theres a lot of
    situations to go through where I might never see a good person come
    into being.

 Shes  going to be taking a lot of risks for the show. But already in
    getting it started, what Ive learned is Ill be the person whos gonna
    change her career, and become a great person rather than I was ready
    to see.

Do your thoughts have that stuck as part of any of the steps that come
    with her (saying for Regina) into it?

Thanks to the producers, thats a guide, even if it was in September. But
    I dont know, from my conversations with the show, it still hasnt
    happened, but it is something I know I want for my life, I want to

        get more done on TV. I dont necessarily script, in some acting, but I
         think I have heard that.

You have family wishes? What other plans? When he comes to make a movie,
    I have to think I dont know about people like that, but I think when
    shes around, itll be to see if Elsa responds to what she feels like.
    If she feels she needs something, I have to think about something,
    like if shell want to make a film. What<|endoftext|>

