# OpenReview forum: "Geometry-aware Euclidean Diffusion Language Generation"
_ICLR.cc/2026/Conference — Submitted to ICLR 2026_

### Official Review · Reviewer_3KAZ · 2025-10-30

**Soundness:** 2
**Presentation:** 1
**Contribution:** 2
**Rating:** 2
**Confidence:** 3

**Summary:**

This paper introduces a variation of diffusion modeling for natural language.
To leverage the strengths of continuous diffusion models, the authors propose techniques to apply these models to inherently discrete language data.
A continuous projection operator, defined by an embedding dictionary, is used to map discrete data into a continuous space, and this projection is incorporated into the formulation of the diffusion ELBO loss.
Additionally, a relaxed version of the ELBO is introduced to enable optimization via standard gradient backpropagation.
Furthermore, to complement the conventional Perplexity metric, the authors introduce the Perceptual Score---a metric that uses an LLM as a proxy for human evaluation.
Experiments are conducted on the LM1B and OpenWebText datasets, and the proposed methods are compared to existing approaches.

**Strengths:**

1. The paper tackles the important topic of designing diffusion models for discrete data such as natural language, and explores new approaches to data geometry and modeling in both discrete and continuous domains.
2. The proposed method, being continuous, has the potential to be combined with consistency modeling and discriminative score distillation, which is a clear advantage as mentioned in Section 4.

**Weaknesses:**

1. Key mathematical symbols and terms are not properly defined, and several equations lack sufficient explanation or correct notation. Examples of unclear variables are as follows:
    * The driving noises B of SDE in Eq. 1, 2, around line 122, and in Table 1.
    * $\mathbf{e}^g$ around line 145.
    * $\tilde{p}$ around line 160.
    * $\mathbf{n}$ in Eq. 14.

2. The derivation and presentation of important equations (e.g., Eq. 4, 5, 13–14) are incomplete or confusing. Providing derivations or relevant references would enhance the clarity of the paper.
    * Regarding the RELBO in Eq. 10, it should be mentioned that the derivation is described in the appendix.

3. The novelty of the proposed method compared to existing discretization approaches is not clearly established.
   In particular, the discussion of data geometry is unconvincing, and the claimed use of Riemannian components, in Table 1, is not clearly demonstrated.

4. The explanation of the ranking metric in Appendix E is insufficient, lacking a clear natural language description and details about LLM prompts.
  Although python scripts for the metric are provided, basic information such as which LLM is used and what prompts are used should, in principle, be described in the paper.
  In addition, it seems necessary to provide evidence for the effectiveness of the proposed metric based on LLMs, such as its correlation with human evaluation metrics.

5. Although perplexity is not a perfect metric, the results for the proposed method in Table 3 are notably poor. It would be helpful to include some discussion on the possible reasons for this and the limitations of perplexity as an evaluation metric regarding this point.

6. In Figure 2, it is unclear why the red dashed line is ideal.

7. There are some typos:
    * in line 173, "can be achieve" should be "can be achieved."
    * unnecessary parentheses around line 100.
    * The explanation of $\epsilon_{\perp}$ and $\epsilon_{\parallel}$ around line 265 is confusing and contains typos.

**Questions:**

1. Regarding Weaknesses 1 and 2, could you elaborate unclear notations and derivations of equations?
2. Regarding Weakness 3: Could you provide further explanation on the difference of the proposed method from previous methods? What is the Riemannian metric considered in the paper?

---

> ### Author Response · Authors · 2025-11-23
> **Response to Reviewer 3KAZ**
>
> [**W-1**] We have revised the paper to give the definition of each variable. Specifically,
>
> (**w-1-1**) $d\mathcal{B}$ denotes the standard Wiener motion in Euclidean space. For the notion in Table 1, we have changed them to the integration results to help you better understand it.
>
> (**w-1-2**)  $\mathbf{e}^g$ indicates the one-hot vector with the number of $g$ classes. We have changed it to $\{0,1\}^g$
>
> (**w-1-3**)  We have removed all $\widetilde{\quad}$ for probability and density functions.
>
> (**w-1-4**) $\mathbf{n}$ indicates a tensor of standard Gaussian noise.
>
> [**W-2**] Here, we further illustrate the meaning of each equation. Moreover, we have added 2 figures (Figs. 1 and 2) to help you get deep understanding of the proposed method.
>
> (**W-2-1**) Eq. (4) renders the global probability transition process from standard Gaussian to discrete probability vector. You can refer to Fig 2, which contains a PDF squeezing process, i.e., the first two sub-figures in Fig 2, the continuous section. Then, it reaches singularities and makes the discretizations as the third sub-figure. Moreover, Eq.(5) can be simply inferred by the score-function formulation of the Gaussian diffusion model.
>
> (**W-2-2**) For Eq. (12), we have added further illustration in Appendix F-1. It illustrates the ratio between the space in the yellow circle of Fig. 2 and the total space. Eq.(13) formulates the variation of SNR(t) within the proposed data geometry re-parameterization framework.
>
>
> [**W-3**] The reviewer must misunderstand some parts of our work. The proposed method is **fundamentally** different from current continuous and discrete diffusion methods. Figs.1 and 2 clearly indicate the novelty of the proposed method. We project the data manifold onto a low-dimensional simplified hyper-sphere and design a continuous diffusion scheme that directly predicts manifold tokens.
>
> [**W-4**] We have added detailed explanation of in the manuscript. We applied the Gemini-2.5-pro for measuring generation quality. We have also listed the prompts in Appendix G1.
>
> [**W-5**] After extended training with an equal number of iterations, our method significantly outperforms all comparison methods. However, we note that Perplexity (PPL) has inherent limitations as a metric, often converging toward trivial solutions. Because PPL is calculated by AR models, the metric is highly sensitive to early tokens; a low-probability assignment in the initial phase creates a poor conditioning context for subsequent tokens. Consequently, the aggregate probability often fails to accurately reflect the actual quality of the generated text.
>
> [**W-6**] Diffusion model is designed to progressively transfer source distribution, e.g., Gaussian, to the target distribution, i.e., data distribution. Thus, intuitively, an ideal transition process needs to progressively increase the SNR, which resulting a red linear line in Fig.2.
>
> [**W-7**] We have corrected those component and added further discussions.

---

> ### Author Response · Authors · 2025-11-24
>
> Hi, Reviewer 3KAZ. We have conducted empirical experiments and provided theoretical explanations to resolve your concerns. Feel free to discuss if you have any further concerns.

---

> ### Comment · Reviewer_3KAZ · 2025-11-26
>
> Thank you for your rebuttal and constructive discussion. I have additional comments and questions.
>
> **Regarding the response to W-1 and notation issues:**
> Thank you for fixing notation issues. However, several notations still require modifications. For example:
> - For improved clarity, it should be explicitly mentioned that $d\tilde{\mathcal{B}}_t$ represents a reversed Brownian motion
> - The tensor $\mathbf{n}$ is used in Table 1, but its definition appears in Line 334. To enhance readability, the definition should be provided before its first appearance.
> - As mentioned by Reviewer j5Rr, In Eq. (4) and when $t=0$, the divergence $\mathcal{D}$ still compares $p_\theta(\mathbf{x}_t)$ and $p(\mathbf{z})$, which are probability distributions on different spaces. I suppose that this issue can be fixed by defining the probability of $\mathcal{P}^{(c)}(\mathbf{z})$ and using this instead of $p(\mathbf{z})$.
>
> **Regarding the response to W-2-1:**
> Thank you for providing Figure 2. Figure 2 illustrates the generative process considered in this paper. However, I do not think Figure 2 explains why Eq. (4) gives the optimization target for the proposed method.
>   - If there is a reference for Eq. (4), could you please show that?
>   - Could you please explain why the bottom row of Eq. (4) (the case where $t \in (0,1)$) is derived?
>     Is it directly derived from Fokker-Planck equation of Eq. (2)?
>   - The two argments, $\frac{d p_\theta (\mathbf{x}_t)}{dt}$ and $-\frac{d(f(\mathbf{x}_t, t)p(\mathbf{x}_t))}{d\mathbf{x}_t} + \frac{1}{2} \frac{d^2( g^2(\mathbf{x}_t, t)p(\mathbf{x}_t))}{d\mathbf{x}_t^2}$, appear to be not probability distributions but scalers. So is the divergence $\mathcal{D}$ in this equation L2 distance?
>
> **Regarding the response to W-6 and the SNR stuffs:**
> Let me elaborate my question as follows:
>   - In the discussion around Figure 4, SNR schedule $\lambda(t)$ such that $\frac{d \lambda}{dt}$ is a constant is regarded as the ideal SNR schedule.
>     However, there are design choices of SNR schedule. For example, Table 1 in [1] shows several schedules of SNR.
>     What is the motivation to use such a linear SNR schedule? Is this crutial for the proposed method?
>   - By the way, I suspect that notations in Section 3.3 still require modifications. For example, the vertical axis indicates $1 - \lambda$ in Figure 4; however, considering Eq. (14) and the definition of $\lambda(t)$ in Line 322, $(1-\lambda)$ should increase as $t$ goes from 0 to 1.
>
> [1] Hang, Tiankai, et al. "Improved noise schedule for diffusion training." Proceedings of the IEEE/CVF International Conference on Computer Vision. 2025.

---

> ### Comment · Reviewer_3KAZ · 2025-11-26
>
> **Regarding the response to W-3:**
> Thank you for presenting Figure 1. After taking look into Figure 1, I reconigze the novel point of the proposed generative process as that the estimated clean data point $\hat{x_\theta}$ at each iteration is ensured to be an exact point of a token embedding by utilizing the projection
> $\mathcal{P}_{\theta}^{(c)}(\mathcal{K}(p_θ(\hat{\mathbf{z}} | \mathbf{x}_t) ))$,
> while Plaid and RDLM admits the estimated clean data point to be interpolation of token embeddings.
>
> However, it raises additional concerns.
>   - Describing the state space for the proposed method as "+Riemann.", in Table 1, would not be accurate. A Riemannian manifold is a smooth manifold equipped with a metric tensor called Riemannian metric [2]. A set of discrete token embeddings, which is paid attention to in the proposed method, is not a smooth manifold. Even when the considered space is extended to the continuous space $ \mathbb{R}^s $, it boils down to a Euclidean space. (Of course, a Euclidean space is a Riemannian manifold such that the Riemannian metric is represented as the identity matrix, though.) Another description would be more suitable.
>   - The reason why $\hat{\mathbf{x}}_\theta (\mathbf{x}_t)$ should belong to the set $ \mathcal{P}^{(c)}(\mathbf{z}) $ of
>  token embeddings is unclear. It is certain that each sampled $\mathbf{x}_0$ should belong to the set of token embeddings.
> However, it does not mean that $\hat{\mathbf{x}}_θ (\mathbf{x}_t)$ should be also an exact token embedding.
> To my understanding, the proposed method is built upon the optimization of Eq. (8) and $ || s_θ(\mathbf{x}_t, t) - \nabla_x \log p(\mathbf{x}_t) ||$ in Eq. (8) is equal to the quantity in Eq.(5).
> When there is no constraint, such as the constraint in Eq.(6), on $\hat{\mathbf{x}}_θ(\mathbf{x}_t)$, minimization of Eq.(8) is achieved when $\hat{\mathbf{x}}_θ(\mathbf{x}_t) = \mathbb{E}[\mathbf{x}_0 | \mathbf{x}_t]$.
>   Namely, the optimal $\hat{\mathbf{x}_θ}(\mathbf{x}_t)$ is the conditional expectation of $\mathbf{x}_0$ given $\mathbf{x}_t$ and thus usually varies from exact token embeddings.
>   In this sense, the constraint in Eq. (6) seems to impose rather suboptimality on the diffusion model framework.
>   For this reason, the motivation of imposing Eq.(6) would be unclear. Provinding further explanation regarding this point would strengthen the motivation of the proposed method.
>
> **Regarding the proposed rank metric:**
> Thank you for providing details of the proposed rank metric. However, validity of the proposed rank is still unclear.
> I believe a validation analysis such as evaluating correlation between human assessment and the proposed metric is required when an automatic evaluation metric is proposed, such as evaluations in [3].
>
> **Layout issues of the revised manuscript:**
> There are several layout issues. For example, Table 3 overlaps with the main text and the margins around some section headings, e.g., "4.1 EXPERIMENTAL COMPARISON ON LM1B", are unreasonably narrow.
>
>
> Best regards.
>
> ---
>
> [2] Lee, John M. Introduction to Riemannian manifolds. Vol. 2. Cham: Springer, 2018.
>
> [3] Lin, Yen-Ting, and Yun-Nung Chen. "LLM-Eval: Unified Multi-Dimensional Automatic Evaluation for Open-Domain Conversations with Large Language Models." Proceedings of the 5th Workshop on NLP for Conversational AI (NLP4ConvAI 2023). 2023.

---

> ### Author Response · Authors · 2025-11-27
> **Response to Reviewer 3KAZ's Additional Comments (Part I)**
>
> Dear Reviewer **3KAZ**
>
> Thanks for your recognition and further comments. We have comprehensively addressed them as follows.
>
> [**Q-1**]
>
> (1) For clarity, we have added widetilde to all reverse Brownian motion terms.
>
> (2) We have added the corresponding definitions for each variable below Table 1: "$\mathbf{x}_0$, $\mathbf{x}_t$, and $\mathbf{n}$ represent the cleaned sample, interpolated state, and Gaussian noise, respectively. $\alpha(t)$ and $\beta(t)$ correspond to their respective weights."
>
> (3) We have corrected this point to $\mathcal{D} _ {0} \left( p _ {\theta} (\hat{\mathbf{z}}|\mathbf{x} _ t), p( \mathbf{z} | \mathbf{x} _ t) \right)$, which represents the KL-divergence between two discrete probability vectors.
>
> [**Q-2**]
>
> (1) This distinction arises from the **different definitions** of the metric. Since we regularize embeddings in a spherical space, effective perturbations are noise components within the tangent space. Consequently, we define the SNR based on an angular metric ranging in $[0,1]$, in contrast to the standard formulation $\frac{\alpha_t}{\sigma_t}$, which ranges in $[0, +\infty)$.
>
> For diffusion models, the SNR is a critical factor governing the denoising rate. This is particularly important for data residing in non-Euclidean spaces; an accurately defined SNR enables the model to learn the exact data generation process. Conversely, a drastically mismatched SNR curve (e.g., the right-upper corner of Fig. 4) prevents the model from mastering the generative process, leading to underfitting on pure noise states without sufficient training with different noised intensity. Moreover, we will release the script to use MCMC for properly selecting parameters.
>
> (2) We have corrected the y-axis to $\lambda$ according to the comment.
>
> [**Q-3**]
>
> We have added further illustrations / corrections of Eq. (4) to solve your 3 concerns.
>
> (1-2) The first end-point regularization ensures the model can make correct discretizations at $t=0$; and second regularization pushes the variation of the probability space for score function to be identical to the variation of the predefined forward diffusion process.
>
> The right part of second regularization, i.e., $- \frac{d(f(\mathbf{x} _ t,t)p(\mathbf{x} _ t))}{d\mathbf{x} _ t}+\frac{1}{2} \frac{d ^ 2(g ^ 2(\mathbf{x} _ t,t)p(\mathbf{x} _ t))}{d\mathbf{x} _ t ^ 2}$ is derived from fokker-planck equation, which is a critical equation to describe the continuous-time probability evolving.
>
> (3)$\mathcal{D} _ 0(\cdot)$ (resp. $\mathcal{D} _ 1(\cdot)$) represents the divergence measure of KL-divergence (resp. MSE);
>
> [**Q-4**]
>
> (1) Actually, the projection process is designed to project the high-dimensional space to a spherical shape, which results in a smooth manifold as the reviewer mentioned;
>
> (2) To accurately trace the true reverse probabilistic flow using a parameterized score function (a fundamental requirement of diffusion models), the predicted state $\hat{\mathbf{x}}_0(\mathbf{x}_t)$ should reside within the valid embedding space $\mathcal{P}^{c}(\mathbf{z})$.
>
> While you note that from an **optimization perspective**, the learned score may **converge to** the expectation of $\mathbf{x} _ 0$, we emphasize that this is merely the result of minimization, not an accurate representation of the generative modeling process for discrete data. Our diffusion scheme evolves a discrete probability distribution $p(\mathbf{z})$ into a Gaussian prior via the transition $\mathbf{x} _ t = \alpha _ t \mathbf{x} _ 0 + \beta _ t \mathbf{n}$. Reversing this specific trajectory requires precise score matching, where the model score $\nabla _ {\mathbf{x} _ t} \log p _ {\theta}(\mathbf{x} _ t) = \frac{\mathbf{x} _ t-\alpha _ t\hat{\mathbf{x}} _ 0}{\beta _ t^2}$ should align identically with the true score $\nabla_{\mathbf{x} _ t} \log p(\mathbf{x} _ t) = \frac{\mathbf{x} _ t-\alpha _ t\mathbf{x} _ 0}{\beta _ t^2}$ (as derived in Eq. (25)).
>
> Moreover, there is no **expectation operation** within the norm of the score-matching objective $\||\nabla _ {\mathbf{x} _ t} \log p _ {\theta}(\mathbf{x} _ t) - \nabla _ {\mathbf{x} _ t} \log p(\mathbf{x} _ t)\|| _ 2$. Therefore, to accurately trace the probabilistic flow, $\hat{\mathbf{x}} _ 0$ should coincide exactly with the discrete ground truth $\mathbf{x} _ 0$ (i.e., the projected dictionary embedding $\mathcal{P}^{(c)}(\mathbf{z})$), rather than a continuous estimation.
>
> Finally, utilizing the expectation $\mathbb{E}[\hat{\mathbf{x}}_0|\mathbf{x}_t]$ introduces another critical issue: **the resulting estimate typically does not coincide with any exact vector in the embedding dictionary**. This mismatch complicates the final discretization step, making it difficult to accurately decode the target token.

---

> ### Author Response · Authors · 2025-11-27
> **Response to Reviewer 3KAZ's Additional Comments (Part II)**
>
> [**Q-5**] According to your comments, we have appended the new assessment of consistency between the human evaluation and the LLM evaluation results in Sec.G.2 of the resubmitted manuscript. Here, we calculate the Spearman's correlation factor $\rho$. Experimental results are shown in the following Table. The proposed LLM-based metric demonstrates a strong correlation with human evaluation, achieving a coefficient of $82.76\%$. Furthermore, we observe that the average human rank score of $0.3971$ aligns closely with the Gemini-based ranking results $0.4$.
>
> | | Gemini | P1 |  P2 |  P3  |  P4 |  P5 |  AVG(P1-5)|
> |  ----  | ----  |  ----  | ----  |  ----  | ----  |  ----  |  ----  |
> | Rank |  0.4        |  0.3857 |  0.4286  |  0.3857 | 0.3714  | 0.4143  |  0.3971|
> | Spearman's $\rho$ | 1.0 |  0.7309  |  0.8839 |  0.7908  |  0.8208 |  0.9116 | 0.8276 |
>
>
>
> [**Q-6**] We have corrected the layout issues.
>
> If you have any further concerns, feel free to discuss. We are glad to resolve any remaining concerns.

---

### Official Review · Reviewer_wMn6 · 2025-10-31

**Soundness:** 3
**Presentation:** 2
**Contribution:** 2
**Rating:** 4
**Confidence:** 3

**Summary:**

This paper proposes a generative framework that models discrete token generation as continuous trajectories of a Gaussian stochastic process within a Euclidean space. To address the challenges posed by high-dimensional discrete data, the authors introduce two core components: a projection function to embed discrete tokens into a continuous domain, and a metric function to infer the conditional probability distribution of subsequent tokens from continuous embeddings.

Furthermore, the paper formulates a novel geometry-aware score that explicitly exploits the inherent manifold structure of discrete language data to refine the fidelity of score approximation. Since direct optimization of the score function is intractable, the authors propose optimizing a tractable surrogate objective, the Relaxed Evidence Lower Bound (RELBO), which ensures a bounded approximation error via continuous relaxation.

Section 3.3 discusses specific model specifications, including the design of the continuous projection function $P^{(c)}(\cdot)$, SDE customization, and the reverse integral form. Empirical validation on the LM1B and OpenWebText benchmarks demonstrates the effectiveness of the proposed framework, showing performance competitive with state-of-the-art models.

**Strengths:**

- **Bridging discrete and continuous domains**: The paper proposes geometry-aware score matching techniques for handling discrete token generation via Euclidean diffusion, achieving Gaussian diffusion language generation that recognizes the inherent structure of discrete language data and provides a theoretically consistent framework.

- **Compatibility with editing and non-monotonic generation**: Unlike auto-regressive models that commit to a fixed prefix, diffusion models can refine the entire sequence representation simultaneously, enabling non-monotonic generation and providing unprecedented control over the generative process, including insertion, editing, and constrained generation.

- **Concrete guidelines for model specifications**: Section 3.3 provides practical implementation guidelines, including theoretical analysis of projection density (Eq. 12), embedding constraints via hypersphere regularization, and SNR curve design.

- **Competitive performance in initial experiments**: On the LM1B dataset, the proposed method achieves a 100% win rate against RDLM and a Ranking score of 0.51. On OpenWebText, with only 17% of training steps, the method achieves comparable results to MDLM and SEDD, and outperforms some auto-regressive Transformer outputs.

- **Potential for future improvements**: Being implemented as a continuous model, the framework can directly leverage mature image diffusion techniques (e.g., consistency training, discriminative score distillation) for further performance enhancements.

**Weaknesses:**

### 1. Concerns regarding the reliability of evaluation metrics

The proposed Perceptual Score (Ranking metric) uses LLMs as proxies for human experts, but lacks sufficient validation of reliability in the following aspects:

- Automatic evaluation by LLMs can be influenced by presentation order and writing style, potentially biasing results in favor of the proposed method
- Details about the LLM model name, prompt design, and evaluation criteria are unclear, making reproducibility verification difficult

**Suggestion**: The robustness of the evaluation should be reinforced through correlation analysis with human evaluation and the use of additional automatic evaluation metrics.

### 2. Lack of empirical evidence for scaling laws

Due to computational constraints, scaling laws with varying model sizes and training data amounts have not been empirically validated. The OpenWebText experiment was completed with only 0.17M steps, leaving the long-term training behavior unclear.

**Suggestion**: Even at small scales, performance trends across different parameter counts should be demonstrated to provide initial evidence of scaling characteristics.

### 3. Absence of ablation studies

While geometry-aware score matching techniques and RELBO are highlighted as main contributions, the individual contributions of each component have not been validated. The loss function consists of cross-entropy and $L_2$ distance terms, but the impact of weighting these terms is unclear.

**Suggestion**: At minimum, comparative experiments should be conducted on: the presence/absence of geometry-aware score, weighting of each loss term, and comparison between RELBO and standard ELBO.

### 4. Insufficient experimental validation of model specifications

Section 3.3 presents detailed theoretical analyses including embedding density analysis, hypersphere regularization, and SNR curve design, but lacks experimental validation of how these design choices contribute to final performance.

**Suggestion**: Experiments are needed comparing: with/without hypersphere regularization, performance across different embedding dimensions, and variations in SNR design.

**Questions:**

### Q1. Response to weaknesses

Could the authors provide their perspectives on the four weaknesses mentioned above? Specifically:

1. **Evaluation metrics**: Validation of bias mitigation strategies for the Ranking metric and plans for using additional evaluation metrics
2. **Scaling**: Plans for conducting small-scale scaling experiments
3. **Ablation study**: Plans for additional experiments to quantify the individual contributions of each component
4. **Model specifications**: Experimental evidence demonstrating that the design choices in Section 3.3 contribute to performance improvements

### Q2. Applicability of image diffusion techniques

The paper states that as a continuous model, the proposed method can directly apply mature image diffusion techniques. However, the data estimator is $P_{\theta}\^{(c)}(\hat{z}_{\theta})$, which appears structurally different from conventional image diffusion models.

**Question**: Despite this architectural difference, can consistency training and distillation techniques be applied without issues? If modifications are necessary, what kinds of adaptations would be required?

---

> ### Author Response · Authors · 2025-11-23
> **Response to Reviewer wMn6**
>
> [**W1**]**Evaluation Metrics**. During the evaluation phase, we have also shuffled the order of the text during evaluation. Moreover, we have attached the evaluation code for your reference.  Note that all models are trained on the same dataset. Thus, it. We leverage the Gemini-2.5-pro as the metric LLM to evaluate their performance.
>
> [**W2**]**Scaling the Training Process**. We have added the experimental comparisons with 1M training steps that show **the proposed geometry-aware score matching consistently surpassing other models with respect to all metrics**. It indicates the superiority of the proposed learning scheme.
>
>
> [**W3-4**]**Ablation Studies**. We emphasize that both the hyperparameters and loss terms are **grounded in theoretical design and analysis**. Regarding the loss function, the MSE term is introduced to regularize the score direction during diffusion, while the KL-divergence term is utilized to enable discrete variable estimation. The absence of either component renders the method ineffective. Consequently, the superior performance of the proposed method serves as empirical validation of these design choices. Due to the significant computational resources required to train additional ablation models, we will **open-source the relevant scripts to assist in hyperparameter selection**. To solve your concern, we have added further ablation studies for those selectable parameters in Sec. 4.3.
>
> [**Q2**]**Applying to Image Diffusion**. The only necessary modification for the proposed method applying to the continuous data, e.g., image data, is an accuracy tokenization method, which can decompose the image data to a large vocabulary with exactly on data manifold tokens. Then, we can generate those tokens with the proposed framework.

---

> ### Author Response · Authors · 2025-11-24
>
> Hi, Reviewer wMn6. We have conducted empirical experiments and provided theoretical explanations to resolve your concerns. Feel free to discuss if you have any further concerns.

---

> > ### Comment · Reviewer_wMn6 · 2025-11-27
> >
> > Thank you for addressing my concerns. As Reviewer j5Rr has pointed out, this paper was submitted in a considerably incomplete state (e.g., main experiments were not completed, and the manuscript contained numerous inaccuracies). Therefore, I believe we should exercise careful judgment regarding the substantial revisions made during this rebuttal period.
> >
> > On the other hand, I agree that the revised content has clearly improved the quality of this paper.
> >
> > **Regarding [Q2], there seems to be a misunderstanding (and I apologize if my phrasing was confusing). What I am asking is not "how to apply this method to image data," but rather "whether mature image diffusion techniques can be applied to this method (for language generation)."**
> >
> > In Section 5, the paper states: "Based on the continuous modeling, we can adapt the mature image diffusion techniques directly onto the language generation field." However, the data estimator in this method includes non-differentiable operations, which creates a structural discrepancy from image diffusion models (which estimate the data itself or the noise).
> >
> > **My concerns are as follows:**
> >
> > 1. Consistency training and distillation typically assume continuous differentiability of the inference process
> > 2. While RELBO-based relaxation is used during training, K(·) is used during inference (Algorithm 2), and this discrepancy may pose problems
> > 3. Can these techniques be applied "directly" even in the presence of this discretization step?
> >
> > If application is possible, could you please explain the theoretical justification or necessary modifications? Alternatively, if you have any preliminary experimental results, I would appreciate if you could share them.

---

> ### Author Response · Authors · 2025-11-28
> **Response to Reviewer wMn6's Additional Comments**
>
> Hi Reviewer wMn6,
>
> Thanks for your clarification. We acknowledge that directly transferring **every component** of standard diffusion to our proposed framework is infeasible. However, specific components—such as solvers (e.g., the Heun predictor-corrector method employed in our approach), remain applicable and effective. In the following section, we outline a detailed illustration for the each term of your concerns.
>
> [**Q1**] **Consistency Model.** Although the inference process involves the non-differentiable operator $\mathcal{K}(\cdot)$, we can employ discrete-time consistency distillation to regularize the diffusion process. Specifically, standard discrete-time consistency distillation enforces consistency on the diffusion model's outputs between two adjacent points on the trajectory, formulated as:
>
> $$\mathcal{L} _ {CD} ^ N(\theta,\theta ^ - ;\phi) = \mathbb{E}\left[ \mathcal{D}(f _ {\theta}(\mathbf{x} _ {t _ {i+1}},t _ {i+1}), f _ {\theta^-}(\hat{\mathbf{x}} _ {t _ {i}},t _ {i})) \right].$$
>
> In our proposed method, we achieve this by regularizing the output of two adjacent points on the trajectories as:
>
> $$\mathcal{L} _ {CD} ^ N(\theta,\theta ^ - ) = \mathbb{E}\left[ \mathcal{H}\left( p _ {\theta}(\hat{\mathbf{z}}|\mathbf{x} _ {t _ {i+1}}), p _ {\theta ^ -}(\hat{\mathbf{z}}|\mathbf{x} _ t)\right) + \|\mathbf{w} p _ {\theta}(\hat{\mathbf{z}}|\mathbf{x} _ {t_{i+1}}) - \mathbf{w} p _ { \theta ^ -}(\hat{\mathbf{z}}|\mathbf{x} _ {t_{i}}) \| _ 2 \right],$$
>
> where $\hat{\mathbf{x}} _ {t _ {i}} = \alpha(t _ i) \hat{\mathbf{x}} _ { \theta }(\mathbf{x} _ {t _ {i+1}})+ \beta(t _ i) \hat{\mathbf{i}}(\mathbf{x} _ {t _ {i+1}}, \hat{\mathbf{x}} _ {\theta}) \vphantom{\int_{0}^{1}}$ represents the result of one-step sampling from $\mathbf{x} _ {t _ {i+1}}$ to $\mathbf{x} _ {t _ {i}}$, which inherently contains $\mathcal{K}(\cdot)$. Through the discrete-time consistency model, we can effectively solve the non-differentiability issues of $\mathcal{K}(\cdot)$.
>
> [**Q2**] **Training Inferent Discrepency**. We acknowledge the discrepancy in the score function utilization between the training and inference processes. However, we emphasize that the RELBO is specifically proposed to bridge this gap. Here, we further illustrate the error bound of the original diffusion process as:
>
> $$-\log p _ {\theta}(\mathbf{x} _ 0) \leq \mathcal{D} _ {KL}(p|p _ {\theta}) = \int_0^T g^2(t) \left\| \nabla _ {\theta}\log p _ {\theta}(\mathbf{x} _ t) -  \nabla\log p(\mathbf{x} _ t) \right\| _ 2 ^ 2 dt + c,$$
>
> where $-\log p_{\theta}(\mathbf{x}_0)$ is the non-negative log-likelihood modeling the probability of the model correctly predicting the ground-truth data, and the right-hand side is the parameterized evidence lower bound (ELBO), which corresponds exactly to the error in the inference process involving $\mathcal{K}(\cdot)$. Moreover, our optimized RELBO acts as an upper bound to the ELBO:
>
> $$\text{RELBO} = \mathbb{E} _ {p} \{ \int _ {0} ^ {1} \frac{d \lambda}{dt} \left[ \left\| \mathbf{w} \mathcal{K}(p _ {\theta}(\hat{\mathbf{z}} | \mathbf{x} _ t)) - \mathbf{w}p _ {\theta}(\hat{\mathbf{z}}|\mathbf{x} _ t) \right\| _ 2 + \left\|\mathbf{w}p _ {\theta}(\hat{\mathbf{z}}|\mathbf{x} _ t) - \mathbf{w}\mathbf{z} \right\| _ 2 \right] dt \} + \mathbb{E} _ {p,p _ {\theta}} \mathcal{D} _ {KL} > \text{ELBO}.$$
>
> Thus, although differences exist between the training and inference processes, we optimize the upper bound of the exact sampling, ensuring the method remains effective and theoretically grounded.
>
>
> [**Q3**] **General Applicability**. Several existing techniques are applicable to our framework, such as the aforementioned discrete-time consistency distillation and various integrators. However, to fully unleash their effectiveness, they require adaptation that accounts for our proposed score-matching mechanism.
>
> If you have any further concerns, feel free to discuss. We are glad to resolve any remaining concerns.

---

### Official Review · Reviewer_EYEn · 2025-11-04

**Soundness:** 3
**Presentation:** 2
**Contribution:** 3
**Rating:** 4
**Confidence:** 2

**Summary:**

The paper introduces a diffusion language model that performs denoising in a continuous embedding space for discrete text tokens. During training, it uses two losses. First, a cross-entropy loss on the logits ($p_\theta(z \mid x_t)$), which are the model outputs before discretization. Second, a MSE loss after snapping (discretization): take the logits, discretize to a one-hot token, then project back to the embedding space with $P^{(c)}$; the MSE matches this re-embedded token to the ground-truth embedding. These losses encourage cycle consistency between the discrete and continuous spaces, i.e., ($z \approx P^{(r)}(P^{(c)}(z))$) and ($x \approx P^{(c)}(P^{(r)}(x))$), where $P^{(c)}$ maps tokens -> embeddings and $P^{(r)}$ maps embeddings -> token distributions.

During inference, it starts from a prior noise state in the embedding space. At each step, the model predicts logits, then discretizes them to a one-hot token. This is either probabilistically (sampling from softmax) or deterministically (argmax). The chosen token is then projected back to the continuous embedding space with $P^{(c)}$ to form the geometry-aware clean estimate used in the reverse diffusion update. After the final step, the method outputs the discrete token sequence.

The method shows competitive performance to the diffusion language model baselines with much fewer training iterations, and surpassing both perceptual quality using the llm-as-judge. However, they have shown that perplexity does not show the good results, as the authors has posit that the confidence of the model is not aligned well with the quality of the prediction.

**Strengths:**

1. The paper explores the possibility of Diffusion LLM that works not in the discrete space but in the continuous space, which better aligns with the original Diffusion setups.

**Weaknesses:**

1. **Relation to DDIM is under-discussed.** Both LanGeo and DDIM (Song et al., ICLR 2021) convert the intermediate latent into a "clean" representation and then re-inject noise to continue the trajectory. The high-level procedure appears conceptually similar, but the paper does not position LanGeo clearly with respect to DDIM or explain the differences.
2. **perplexity vs. perceptual quality claim is weakly supported.** The paper asserts that perplexity is not well-aligned with perceptual quality, but provides limited citations and concrete examples. In practice, if the reference AR model used to compute perplexity is very strong (e.g., Qwen3), the misalignment may be smaller. Moreover, the proposed LLM-as-Judge ranking metric is hard to interpret: a score of 1.0 vs. 0.51 does not imply "2× better", and the comparison is further limited by evaluating against only two models in Table 2, which reduces statistical strength and external validity.
3. The paper does not explicitly specify which model was used as the LLM-as-Judge in the text.

**Questions:**

I am not deeply familiar with diffusion or language modeling, so I will carefully consider other reviewers’ comments before making a final decision. I do not have sufficient understanding to fully evaluate the significance of the proposed method. However, at present, I lean toward rejection, primarily because the evaluation protocol (the perplexity discussion and the ranking metric) does not convincingly reflect the actual performance of the method.

---

> ### Author Response · Authors · 2025-11-23
> **Response to Reviewer EYEn**
>
> [**W1**] **Comparison with DDIM**. Actually, we want to note to the reviewer that we do not claim any contribution to this solver. This part is **only an SDE trajectory parameterization trick** to simplify the framework. Our main contributions are the data geometry-aware score matching and analysis of its correlation with the classic score-matching scheme. Moreover, the proposed solver only converges to DDIM when $\bar{\gamma}(t)=1$. For other selections of $\bar{\gamma}(t)$, the proposed method is different from DDIM.
>
> [**W2**] **Perplexity**. To further solve your concern, we also list the PPL of QWen-3. Notably, the proposed method **outperforms** all discrete diffusion models and even **significantly surpasses** the SOTA autoregressive-based method. Demonstrating the strong potential and a promising direction for language generative modeling.
>
> |  Method   | PPL-QWen-3  | PPL-DeepSeek  |
> |  ----  | ----  | ----  |
> | AR  | 58.94 |87.59 |
> | MDLM-1k  | 76.66 |123.21 |
> | MDLM-5k  | 39.51 | 51.49 |
> | Ours-1k  | 27.72 | 39.94 |
> | Ours-5k  | 18.24 | 25.92 |
>
> Note that the ranking-like metric is a widely adopted metric, especially in several LLM arenas \& leaderboards (e.g., \url{https://lmarena.ai/}). The winning rate of random anonymous competitions is the most direct method to evaluate the quality of generation. Furthermore, we leverage this ranking metric also because currently advanced LLMs tend to be closed-source, which makes it hard to derive the logits to evaluate the PPL (e.g., the utilized Gemini).
>
> [**W3**]  **Metric LLM**. We mainly leveraged Gemini-2.5-pro as the judge-LLM to compare the generation performance. Moreover, we have also evaluated the results with Gork-4 with the similar results.

---

> ### Author Response · Authors · 2025-11-24
>
> Hi, Reviewer EYEn. We have conducted empirical experiments and provided theoretical explanations to resolve your concerns. Feel free to discuss if you have any further concerns.

---

> ### Author Response · Authors · 2025-11-27
> **Looking forward to your further feedback**
>
> Dear Reviewer **EYEn**
>
> Thank you for taking the time to review our manuscript. We have carefully addressed all the comments and concerns raised to improve the manuscript, as reflected in our detailed responses and the revised manuscript and supplementary material.
>
> We are looking forward to your further feedback.
>
> Best regards,
>
> The Authors

---

### Official Review · Reviewer_j5Rr · 2025-11-06

**Soundness:** 1
**Presentation:** 1
**Contribution:** 2
**Rating:** 0
**Confidence:** 3

**Summary:**

This paper studies diffusion language models, with continuous-state diffusion modeling on latent space. The proposed RELBO (relaxed ELBO) with triangle inequality realizes a differentiable loss on continuous-to-discrete transition at the end of inference. The authors also propose a way to determine the hyperparameters of continuous diffusion through the lens of signal-to-noise ratio. The proposed method is tested against LM1B and OpenWebText datasets, and shows a competitive performance among the continuous-state discrete diffusion models.

**Strengths:**

The authors point out that the current diffusion language models ignore the geometry of discrete data, which is an important point. To bridge this gap, the authors introduce a (perturbed) embedding and run a diffusion model over the embedding space, together with a relaxed objective (RELBO) with better regularity. This idea, with additional polishing, may lead to stronger diffusion language modeling.

**Weaknesses:**

My impression is that the draft is still not completed. There are so many typos and undedined math symbols, and even the main experiment is not finished. I suggest a thorough revision and submission to another venue. Let me mention some points that need improvement and/or clarification.
- [W1] **Incomplete experiment:** First of all, the main experiment on OpenWebText is not finished. I understand that the training requires heavy computation, but "we have only trained the model with only 0.17M steps on OpenWebText by the submission deadline. We will report a fully trained results after finishing the training." in L345 is simply an unfair use of the rebuttal revision. If the authors are going to update the "main result" during rebuttals, it is essentially an extention of submission deadline.
- [W2] **Inaccurate/incomplete presentations:** There are many expressions that are incorrect, ambiguous, or undefined.
    - [W2-1] The two equations in L91 for the solution of SDE should be valid only for a specific class of $f, g$.
    - [W2-2] In Equation 4, $L_t$ is comparing $\tilde{p}$ and $p$, which are probability distributions on different spaces (as they specify $D$ as KL in L160). I believe it requires correction.
    - [W2-3] For the derivation of Equation 11, while I checked the Appendix A.2, apprently it is not completed and L781-L787 does not make sense to me. For instance, what is $Y_t$ in L782? The sentence in L784 seems incomplete, and I do not see how Equation 32 is derived.
    - [W2-4] There are many math symbols without clear definitions in the paper, (at least) as listed below. I found the definitions of some of them in the appendix, but they are not readily available in the main body. (I might have just missed some definitions in the main body though.)
        - L146: What is $e^g$?
        - L153: What is $\tilde{p}$? Is this "$\to$" a definition of $\tilde{p}$?
        - L246: In which sense is "$n$" the diffusion dimension (single token or whole input or..)? What is its relation to $g$, $s$, $L$, which were introduced in Section 3.1?
        - L269: What is $n$ here?
        - L277: What is $O(\cdot)$ in Algorithm 2?
        - L280: What is $\hat{z}_\theta$ in Algorithm 1? What is $E$?
        - L307: What is $\tilde{n}$?
        - L312: What is $\omega$?
        - L782: What is $Y_t$?
    - [W2-5] There are also many typos or English errors, or even incomplete sentences. I strongly recommend some form of proofreading.
        - L54: some some -> some
        - L55: remanian -> Riemannian
        - L80: we extensive experiments -> we conduct extensive experiments
        - L89: $dB$ -> $dB_t$? Many of the time variables in SDE-like equations are missing; is there a reason?
        - L100: "masked diffusion models ()" what's inside the parentheses?
        - L101: transfer -> transfers
        - L120: "are designed based to directly synthesis the , e.g.,": This sentence looks broken.
        - L136: "the another" ??
        - L149: Is it $R^s$ or $R^L$? $P_r$ or $P^{(r)}$?
        - L168: What do the authors want to mean by $\{P(z)\}$? The set as it is a singleton set. I guess something like $\{P(z)\mid z\in Z\}$ was intended?
        - L174: achieve -> achieved
        - L175: determinists -> deterministic
        - L184: boundary -> bound
        - L192: A period is missing.
        - L197, L206: $dt$ is missing.
        - L218: before sections -> previous sections
        - L252: "Due to we regularize the $\hat{x}$ onto a hyper-sphere.": I don't think this is complete as a sentence..
        - L256: $\varphi$ indicate -> $\varphi$ indicates
        - L265: fix "$bm\epsilon$"
        - L265: sensible -> sensitive?
        - L288: exist -> existence
        - L312: $logp$ -> $\log p$

While the proposed idea of relaxed loss computation for discrete-continuous diffusion language modeling is interesting, I believe the current paper is not ready for a review process.

**Questions:**

I have the following questions.
- [Q1] L148: How can it be an invertible function with different dimensions? Do the authors assume a subset of Euclidean spaces in the input? If so, specify the set.
- [Q2] L235: What is "the domain size in a single dimension"? Are the authors talking about hypercube-like objects?
- [Q3] L239: What is "the occpancy range for a single class"? How is it defined mathematically?
- [Q4] L357: I have briefly read the ranking script in the supplementary material, but I wonder why the ranking takes values lower than zero, because in the script the ranking is defined as one-based indexing (so positive integers?). Or are they projected to the $[0, 1]$ interval afterwards?

The following are minor questions. I do not need answers for them but please incorporate them into future revision:
- [Q5] L172: What is "Kronecker delta transition function": it seems the authors itemize some choices of $K$ in L174, none of which is familiar with what I know as "Kronecker delta". Does it just mean a one-hot vector?
- [Q6] L187 etc: Can the authors add parentheses for the expectation operation for readability?
- [Q7] L219: "infinite classes"? "$N$ classes with arbitrarily large $N$" and "$\infty$ classes" are different; I think the authors consider the former?

---

> ### Author Response · Authors · 2025-11-23
> **Response to Reviewer j5Rr**
>
> [**W1**] **Incomplete Experiment**. Due to the very heavy computational resource consumption, our method was only trained 0.17M steps. However, with further training, our method without any further augmentations, e.g., consistency training or trajectory distillation, **can generate results surpassing those of the AR model**, which strongly supports continuously modeling for language data. Due to the **very rapid development of this field**, our submission want to call more attention to the research on continuous diffusion language generation.
>
> [**W2**] **Inaccurate/incomplete presentations.**
>
> [**W2-1**] We have revised this part to add further constraints  for functions $f(\cdot)$ and $g(\cdot)$ to the solvability of the analytic solution.
>
> [**W2-2**] We have transferred $p ( \mathbf{z} )$ to embedding space that $p(\mathbf{w}\mathbf{z})$ to unify the space of $\widetilde{p}$ and $p$.
>
>
> [**W2-3**] Derivation of Equation 11. We have added further elaborations in Appendix A.2. For your convenience, we copy it as the following.
>
>
> $$
>     \mathbb{E} _ {p} \int _ {0} ^ {1} \frac{d\lambda}{dt} \left[ \left\| \mathbf{w}\mathcal{K}(p _ {\theta}(\hat{\mathbf{z}}|\mathbf{x} _ t)) - \mathbf{w}p _ {\theta}(\hat{\mathbf{z}}|\mathbf{x} _ t) \right\| _ 2 + \left\|\mathbf{w}p _ {\theta}(\hat{\mathbf{z}}|\mathbf{x} _ t)- \mathbf{w}\mathbf{z} \right\| _ 2\right] \\
>     \simeq \mathbb{E} _ {(X _ t,Y _ t) \sim p(X _ t,Y _ t)} \frac{d\lambda}{dt} \left( \mathcal{H}(p _ {\theta}(\hat{\mathbf{z}}|\mathbf{x} _ t)) + \| \mathbf{w}p _ {\theta}(\hat{\mathbf{z}}|\mathbf{x} _ t)- \mathbf{w}\mathbf{z} \| _ 2 \right).
> $$
>
> Here, we made such approximation due to that minimizing $\left\| \mathbf{w}\mathcal{K}(p _ {\theta}(\hat{\mathbf{z}}|\mathbf{x} _ t)) - \mathbf{w}p _ {\theta}(\hat{\mathbf{z}}|\mathbf{x} _ t) \right\| _ 2$ can be achieved via minimize the entropy of $p _ {\theta}(\hat{\mathbf{z}}|\mathbf{x} _ t)$, i.e., $\mathcal{H}(p _ {\theta}(\hat{\mathbf{z}}|\mathbf{x} _ t))$. Furthermore, to minimize $\mathcal{H}(p _ {\theta}(\hat{\mathbf{z}}|\mathbf{x} _ t))$, we propose to regularize the divergence between $p _ {\theta}(\hat{\mathbf{z}}|\mathbf{x} _ t)$ and a low entropy target, i.e., $p(\mathbf{z})$ ($\mathcal{H}(p(\mathbf{z})) = 0$). Then, we can result in the following loss term of
>
> $$
>     \mathcal{L} = \mathbb{E} _ {(\mathbf{z},\mathbf{x})\sim p(\mathbf{z},\mathbf{x})} \left[\mathcal{D} _ {KL}\left(p _ {\theta}(\hat{\mathbf{z}}|\mathbf{x} _ t),p(\mathbf{z})\right) + \|\hat{\mathbf{z}} _ {\theta}(\mathbf{x} _ t)\mathbf{E} - \mathbf{z}\mathbf{E} \| _ 2\right],
> $$
>
> [**W2-4**] 2-4-1.  $e^g$ for a g-dimension one-hot vector. We have changed it to {0,1 } $ ^ g$.
>
> 2-4-2.  We have replaced all $\widetilde{p} _ {\theta}$ with $p _ {\theta}$, where $\widetilde{p}$ was introduced to illustrate as a learnable parameterized probability space. Both $p _ {\theta}$ and $p$
>
> 2-4-3.  Here $n$ is equal to $s$.
>
> 2-4-4. $\mathbf{n}$ denotes a standard Gaussian.
>
> 2-4-5. We have corrected $\mathcal{O}(\cdot)$ to $\mathcal{K}(\cdot)$, which indicates the sampling function of a one-hot vector.
>
> 2-4-6. $\hat{\mathbf{z}} _ {\theta}$ denotes the sampled one-hot vector from $p _ {\theta}(\hat{\mathbf{z}}|\mathbf{x} _ t)$. $\mathbf{E}$ has been corrected to $\mathbf{w}$.
>
> 2-4-7. $\widetilde{n}$ represents a resampled standard Gaussian noise.
>
> 2-4-8. $d \mathbf{\omega}$ is equal to $d\mathcal{B}$, which represents the standard Wiener in a stochastic process. We have unified them together.
>
> 2-4-9. $Y _ t$ is $\mathbf{z}$.
>
> [**W2-5**] We have corrected the typos according to your comments (2-5.1, 2, 3, 6, 7, 8, 11, 12, 13, 14, 15, and 16). Here, we also list explanations for the other questions as the following.
>
> W2-5-4. Due to that $d\mathcal{B}$ indicating the **standard** Wiener process, we can abbreviate the $t$ in $d\mathcal{B} _ t$ to make the expression concise.
>
> W2-5-5. Here it is for a reference.
>
> W2-5-9. We have unified the notions of diffusion space to $\mathbb{R} ^ {S}$ and projection function to $\mathcal{P} ^ {(c)}(\cdot)$.
>
> W2-5-10. The $p (\mathbf{z})$ represents a probabilistic vector (one-hot), indicating which $\mathbf{z}$ is being sampled out of the total number of N classes.

---

> ### Author Response · Authors · 2025-11-23
> **Response to Reviewer j5Rr (Part II)**
>
> [**Q1**] **Projection between dimensionality.** Such invertible projection only works for continuous embedding in the set of embedding dictionary, we will add this description in the manuscript.   Because we are projecting between the discrete and continuous variables, there is no restriction for keeping the same dimension.
>
> [**Q2**] **Domain Size**. Yes, exactly, we can approximate the size of a certain dimension via the range between the largest and smallest values from the embedding dictionary.
>
>
> [**Q3**] **Token Occupancy Size**. We define occupancy $\delta _ d$ as the maximum variation around a specific class embedding within error tolerance.  Through introducing a tolerance upper bound, e.g., $\mathcal{E}$, we can formulate the target size $\delta _ d$ as
>
> $$
> \delta _ d =   \arg \min _ {\delta} |\| F _ {\theta}(\mathbf{x} _ t+\delta) - F _ {\theta}(\mathbf{x} _ t) \| _ 2 -\mathcal{E} |.
> $$
>
> Due to that $\mathbf{x} _ t$ actually evolves on the predefined trajectories, i.e., $\mathbf{x} _ t = \sqrt{1-t}\mathbf{x} _ 0 + \sqrt{t} \mathbf{n}$. We can further approximate $\delta _ d$ as time interval $\delta _ t$ to ensure that $\delta _ d \approx \mathbf{x} _ {\delta _ t} - \mathbf{x} _ 0$. Then, we can explore the space defined by the $\delta _ t$, using MCMC to sample over different $t$.
>
> To get the detailed values, we have conducted further experiments using MCMC sampling to explore the correlation between $\delta _ t$ and estimation accuracy. Experimental results are shown in the following Fig. F-1. The accuracy can maintain a very high level as $\delta _ t<2e-2$. According to the aforementioned $\mathbf{x} _ t$ parameterization, the corresponding std of the Gaussian component is $\sqrt{2e-2}$. The occupancy of a single token in $6-dim$ embedding space can be further approximated as $V _ 6(r=3\sqrt{2e-2}) =\frac{\pi^{3}}{6!} \times (3\sqrt{2e-2})^6 \approx8.5e^{-4}$. Considering that the vocabulary contains 50257 tokens, the total occupancy as illustrated by $N\delta _ d^N$ in Eq.~(12) is $8.5e^{-4} \times 50027 = 42.593$ (in a normalized space).
>
>
> [**Q4**] The raw results are in the range of $[1,2]$. We manually normalize the results to the range of $[0,1]$ by subtracting 1 from all results.
>
>
> [**Q5**] The ``Kronecker delta transition function" denotes a sampling process of discrete variables from its probabilistic distribution, resulting in a one-hot vector.  As illustrated in the manuscript, it can be achieved by greedy decoding or probabilistic sampling. Because the probabilistic vector $p _ {\theta}(\mathbf{z})$ is with the same shape of $\mathbf{z}$, this process can be achieved by a Kronecker delta function that transfers the value of a certain position to $1$ and the remaining to $0$.
>
>
> [**Q6**] We have added parentheses to all expectation expressions for your convenience.
>
>
> [**Q7**] We have corrected the expression to ``as $N$ approaching infinity." The number of classes cannot be infinity, we just limit it to approaching infinity.

---

> ### Author Response · Authors · 2025-11-24
>
> Hi, Reviewer j5Rr. We have conducted empirical experiments and provided theoretical explanations to resolve your concerns. Feel free to discuss if you have any further concerns.

---

> ### Comment · Reviewer_j5Rr · 2025-11-26
>
> Thank you for the rebuttal. I cannot readily check if the all the corrections/definitions are now incorporated since it seems difficult to track the changes (partially due to the openreview not showing previous versions), but I believe the authors in that regard.
>
> However [W1] and [W2-1] still persist:
> - [W1] I do not think the initial submission that essentially extends the deadline for the main experiment is appropriate. Particularly, two out of three experimental results in Section 4 are given during rebuttal, which is beyond the role of rebuttal as a clarification or supplementary.
> - [W2-1] My point is that, to obtain the explicit form of the SDE solution, the coefficient functions need to satisfy additional constraints such as being linear etc. The current text describes $f$ as a general function, which is mathematically incorrect because a general non-linear drift would result in a non-Gaussian transition kernel.

---

> > ### Author Response · Authors · 2025-11-26
> >
> > Hi Reviewer j5Rr,
> >
> > (1) In the previously uploaded version, major revisions were marked in cyan. For your convenience, we have updated the manuscript to highlight **all revised content** in magenta, making it easier to distinguish the latest changes.
> >
> > (2) Regarding the rebuttal phase updates, we have focused **solely on adding experiments** to further validate our proposed method; the core methodology remains unchanged. We have strictly followed the ICLR guidelines, which encourage authors to include additional experiments and results. As stated in the guidelines: "...During the discussion/rebuttal phase and for the camera ready, the page limit will be increased to 10 pages **to allow for new results/discussions.** ..." it even allows multiple resubmission.
> >
> > (3) Regarding Section 2 ("Continuous Data Synthesis via Euclidean Diffusion"), we follow the standard formulation established by Song et al. (2020b). in this domain, the forward process is defined as a diffusion process that gradually perturbs data towards a prior (typically Gaussian). The drift coefficient $f(x_t, t)$ in these frameworks (e.g., VP-SDE, VE-SDE) is inherently chosen to be affine to ensure the transition kernel $q(x_t|x_0)$ remains Gaussian. Therefore, our statement that "such a forward process has an analytical solution" refers specifically to this class of forward processes used in diffusion models, rather than arbitrary non-linear SDEs. This is standard terminology in the field.To further address your concern, we have **added the corresponding regularization** for the drift and diffusion functions in the revised manuscript.

---

> > > ### Comment · Reviewer_j5Rr · 2025-11-26
> > >
> > > Thank you for the reply and highlighting the modifications. While the authors are right in that they are allowed as many revisions as they want, it is written in the ICLR guidance that
> > >
> > > > Area chairs and reviewers reserve the right to ignore changes that are significantly different from the original paper.
> > >
> > > Regardless of the revision's quality, I believe that the change in manuscript has been well beyond an acceptable degree for rebuttal revision. In my perspective, [W1] would be significant enough, but it is also clear by looking at the amount of modifications, which is now visible in color. So I will keep my score.
> > >
> > > This said, it is just one reviewer's opinion and I acknowledge that the revision indeed has clarified some parts. Thank you.

---

### Author Response · Authors · 2025-11-23
**General Response to Reviewers**

Thanks for the reviews. We have further polished the writing and presentation of the manuscript based on the comments and uploaded it. Moreover, we want to note that our method, only a plain diffusion framework, with  60 %  winning rate and reducing  70 %  PPL, **significantly outperforms** autoregressive model and the publicly available SOTA method of the discrete diffusion model MDLM on the OpenWebText dataset. Furthermore, given its compatibility with advanced techniques such as consistency training and distillation, **we believe our data geometry-aware score matching represents a highly promising direction with strong potential for natural language modeling!** Main revisions are marked with the color of Cyan in the revised manuscript.

---

### Author Response · Authors · 2025-11-25

Hi Reviewers,

We would like to gently remind you that we have posted detailed responses to your initial comments. We have made a sincere effort to address the concerns raised, specifically regarding the investigation of token occupancy (Appendix A2), the derivation of RELBO (Appendix C), and additional ablation studies (Sec. 4.3), etc. We have updated the manuscript accordingly. As the discussion period is drawing to a close, we would greatly appreciate your feedback to ensure these revisions have fully resolved your concerns. Moreover, we are confident that the **superior performance** of the proposed method indicates a **promising** direction for language modeling.

The authors.

---

### Meta-Review · Area_Chair_Z6wj · 2026-01-06

**Summary:**

This paper proposes a continuous diffusion framework for language generation that models discrete token generation as continuous trajectories in Euclidean space. The method introduces a projection function to embed discrete tokens into continuous space, a metric function for inferring token probabilities, and a novel "data geometry-aware score" that exploits manifold structure. The authors propose optimizing a Relaxed Evidence Lower Bound (RELBO) and introduce a comprehension score metric using LLMs as judges.

The reviews reveal serious concerns about submission completeness, presentation quality, and theoretical rigor, with scores ranging from 0 to 4 (average: 2.5/10).

**Reviewer Concerns:**

Concerns Partially Addressed:
- Missing experiments (j5Rr, wMn6): Authors completed the OpenWebText experiments during rebuttal, training to 1M steps and demonstrating competitive results (60% win rate vs. AR, significantly lower perplexity than MDLM). However, Reviewer j5Rr maintains this constitutes an "unfair extension of the deadline".
- Notation and presentation issues (all reviewers): Authors made extensive corrections to undefined symbols ($B$, $p_θ$, $\tilde{z}$, $\epsilon$), fixed typos, and added clarifications. Reviewer wMn6 acknowledged "the revised content has clearly improved the quality of this paper." However, Reviewer 3KAZ notes that "several notations still require modifications" even after revision, including issues with probability distributions on different spaces (Eq. 4) and reversed Brownian motion notation.
- Evaluation metric validity (EYEn, wMn6): Authors added human evaluation correlation analysis showing Spearman's $\rho = 0.8276$ between Gemini-based rankings and human assessment, provided additional perplexity comparisons with Qwen-3 and DeepSeek, and clarified use of Gemini-2.5-pro as judge. However, fundamental concerns about the ranking metric's interpretability remain.
- Ablation studies (wMn6): Authors added ablation studies in Section 4.3 during rebuttal, but acknowledge that some components (MSE term, KL-divergence term) cannot be ablated as "the absence of either component renders the method ineffective."

Outstanding Critical Concerns:
- Submission integrity and formatting violations (j5Rr):
The most serious issue is that the main experiment on OpenWebText was explicitly incomplete at submission (acknowledged by authors: "we have only trained the model with only 0.17M steps...We will report a fully trained results after finishing the training"). Reviewer j5Rr correctly identifies this as an unfair deadline extension, by stating "the change in manuscript has been well beyond an acceptable degree for rebuttal revision". Combined with formatting violations (overlapping text, manipulated line spacing) in the initial submission, this raises a serious procedural concern.
- Persistent theoretical issues (3KAZ):
Despite extensive revisions, Reviewer 3KAZ (who engaged most deeply on technical content) identifies fundamental problems: 1) The Riemannian manifold framing is mathematically questionable; and 2) the motivation for key constraints (Eq. 6) imposing "suboptimality" is unexplained.

**Reviewer Scores:**

- Reviewer j5Rr: Remains at 0-2/10.
- Reviewer EYEn: Likely remains at 4/10.
- Reviewer wMn6: Likely 4/10.
- Reviewer 3KAZ: Likely remains at 2-4/10.

---

### Decision · Program_Chairs · 2026-01-26

Reject